# USING STOCHASTIC GRADIENT DESCENT TO SMOOTH NONCONVEX FUNCTIONS: ANALYSIS OF IMPLICIT GRADUATED OPTIMIZATION

## ABSTRACT

The graduated optimization approach is a method for finding global optimal solutions for nonconvex functions by using a function smoothing operation with stochastic noise. We show that stochastic noise in stochastic gradient descent (SGD) has the effect of smoothing the objective function, the degree of which is determined by the learning rate, batch size, and variance of the stochastic gradient. Using this finding, we propose and analyze a new graduated optimization algorithm that varies the degree of smoothing by varying the learning rate and batch size, and provide experimental results on image classification tasks with ResNets that support our theoretical findings. We further show that there is an interesting correlation between the degree of smoothing by SGD's stochastic noise, the well-studied "sharpness" indicator, and the generalization performance of the model.

## 1 INTRODUCTION

### 1.1 BACKGROUND

The amazing success of deep neural networks (DNN) in recent years has been based on optimization by stochastic gradient descent (SGD) (Robbins & Monro, 1951) and its variants, such as Adam (Kingma & Ba, 2015). These methods have been widely studied for their convergence (Moulines & Bach, 2011; Needell et al., 2014) (Fehrman et al., 2020; Bottou et al., 2018; Scaman & Malherbe, 2020; Loizou et al., 2021; Zaheer et al., 2018; Zou et al., 2019; Chen et al., 2019; Zhou et al., 2020; Chen et al., 2021; Iiduka, 2022) and stability (Hardt et al., 2016; Lin et al., 2016; Mou et al., 2018; He et al., 2019) in nonconvex optimization.

SGD updates the parameters as $\boldsymbol{x}_{t+1} := \boldsymbol{x}_t - \eta \nabla f_{\mathcal{S}_t}(\boldsymbol{x}_t)$, where $\eta$ is the learning rate and $\nabla f_{\mathcal{S}_t}$ is the stochastic gradient estimated from the full gradient $\nabla f$ using a mini-batch $\mathcal{S}_t$. Therefore, there is only an $\boldsymbol{\omega}_t := \nabla f_{\mathcal{S}_t}(\boldsymbol{x}_t) - \nabla f(\boldsymbol{x}_t)$ difference between the search direction of SGD and the true steepest descent direction. Some studies claim that it is crucial in nonconvex optimization. For example, it has been proven that noise helps the algorithm to escape local minima (Ge et al., 2015; Jin et al., 2017; Daneshmand et al., 2018; Vardhan & Stich, 2021), achieve better generalization (Hardt et al., 2016; Mou et al., 2018), and find a local minimum with a small loss value in polynomial time under some assumptions (Zhang et al., 2017). Several studies have also shown that performance can be improved by adding artificial noise to gradient descent (GD) (Ge et al., 2015; Zhou et al., 2019; Jin et al., 2021; Orvieto et al., 2022).

(Kleinberg et al., 2018) also suggests that noise smoothes the objective function. Here, at time $t$, let $\boldsymbol{y}_t$ be the parameter updated by GD and $\boldsymbol{x}_{t+1}$ be the parameter updated by SGD, i.e.,

$$\boldsymbol{y}_t := \boldsymbol{x}_t - \eta \nabla f(\boldsymbol{x}_t), \ \ \boldsymbol{x}_{t+1} := \boldsymbol{x}_t - \eta \nabla f_{\mathcal{S}_t}(\boldsymbol{x}_t) = \boldsymbol{x}_t - \eta(\nabla f(\boldsymbol{x}_t) + \boldsymbol{\omega}_t).$$

Then, we obtain the following update rule for the sequence $\{\boldsymbol{y}_t\}$,

$$\mathbb{E}_{\boldsymbol{\omega}_t}[\boldsymbol{y}_{t+1}] = \mathbb{E}_{\boldsymbol{\omega}_t}[\boldsymbol{y}_t] - \eta \nabla \mathbb{E}_{\boldsymbol{\omega}_t}[f(\boldsymbol{y}_t - \eta \boldsymbol{\omega}_t)], \tag{1}$$

where $f$ is Lipschitz continuous and differentiable. Therefore, if we define a new function $\hat{f}(\boldsymbol{y}_t) := \mathbb{E}_{\boldsymbol{\omega}_t}[f(\boldsymbol{y}_t - \eta \boldsymbol{\omega}_t)]$, $\hat{f}$ can be smoothed by convolving $f$ with noise (see Definition 2.1, also (Wu,

1996)), and its parameters $\boldsymbol{y}_t$ can approximately be viewed as being updated by using the gradient descent to minimize $\hat{f}$. In other words, simply using SGD with a mini-batch smoothes the function to some extent and may enable escapes from local minima. (The derivation of equation (1) is in Section A.)

**Graduated Optimization.** Graduated optimization is one of the global optimization methods that search for the global optimal solution of difficult multimodal optimization problems. The method generates a sequence of simplified optimization problems that gradually approach the original problem through different levels of local smoothing operations. It solves the easiest simplified problem first, as the easiest simplification should have nice properties such as convexity or strong convexity; after that, it uses that solution as the initial point for solving the second-simplest problem, then the second solution as the initial point for solving the third-simplest problem and so on, as it attempts to escape from local optimal solutions of the original problem and reach a global optimal solution.

This idea first appeared in the form of graduated non-convexity (GNC) by (Blake & Zisserman, 1987) and has since been studied in the field of computer vision for many years. Similar early approaches can be found in (Witkin et al., 1987) and (Yuille, 1989). Moreover, the same concept has appeared in the fields of numerical analysis (Allgower & Georg, 1990) and optimization (Rose et al., 1990; Wu, 1996). Over the past 25 years, graduated optimization has been successfully applied to many tasks in computer vision, such as early vision (Black & Rangarajan, 1996), image denoising (Nikolova et al., 2010), optical flow (Sun et al., 2010; Brox & Malik, 2011), dense correspondence of images (Kim et al., 2013), and robust estimation (Yang et al., 2020; Antonante et al., 2022; Peng et al., 2023). In addition, it has been applied to certain tasks in machine learning, such as semi-supervised learning (Chapelle et al., 2006; Sindhwani et al., 2006; Chapelle et al., 2008), unsupervised learning (Smith & Eisner, 2004), and ranking (Chapelle & Wu, 2010). Moreover, score-based generative models (Song & Ermon, 2019; Song et al., 2021b) and diffusion models (Sohl-Dickstein et al., 2015; Ho et al., 2020; Song et al., 2021a; Rombach et al., 2022), which are currently state-of-the-art generative models, implicitly use the techniques of graduated optimization. A comprehensive survey on the graduated optimization approach can be found in (Mobahi & Fisher III, 2015b).

Several previous studies have theoretically analyzed the graduated optimization algorithm. (Mobahi & Fisher III, 2015a) performed the first theoretical analysis, but they did not provide a practical algorithm. (Hazan et al., 2016) defined a family of nonconvex functions satisfying certain conditions, called $\sigma$-nice, and proposed a first-order algorithm based on graduated optimization. In addition, they studied the convergence and convergence rate of their algorithm to a global optimal solution for $\sigma$-nice functions. (Iwakiri et al., 2022) proposed a single-loop method that simultaneously updates the variable that defines the noise level and the parameters of the problem and analyzed its convergence. (Li et al., 2023) analyzed graduated optimization based on a special smoothing operation. Note that (Duchi et al., 2012) pioneered the theoretical analysis of optimizers using Gaussian smoothing operations for nonsmooth convex optimization problems. Their method of optimizing with decreasing noise level is truly a graduated optimization approach.

## 1.2 MOTIVATION

Equation (1) indicates that SGD smoothes the objective function (Kleinberg et al., 2018), but it is not clear to what extent the function is smoothed or what factors are involved in the smoothing. Therefore, we decided to clarify these aspects and identify what parameters contribute to the smoothing. Also, once it is known what parameters of SGD contribute to smoothing, an implicit graduated optimization can be achieved by varying the parameters so that the noise level is reduced gradually. Our goal was thus to construct an implicit graduated optimization framework using the smoothing properties of SGD to achieve global optimization of deep neural networks.

## 1.3 CONTRIBUTIONS

**1. SGD's Smoothing Property (Section 3).** We show that the degree of smoothing $\delta$ provided by SGD's stochastic noise depends on the quantity $\delta = \frac{\eta C}{\sqrt{b}}$, where $\eta$ is the learning rate, $b$ is the batch size, and $C^2$ is the variance of the stochastic gradient (see Assumption 2.1). Accordingly, the smaller the batch size $b$ is and the larger the learning rate $\eta$ is, the smoother the function becomes (see Figure 1). This finding provides a theoretical explanation for several experimental observations. For

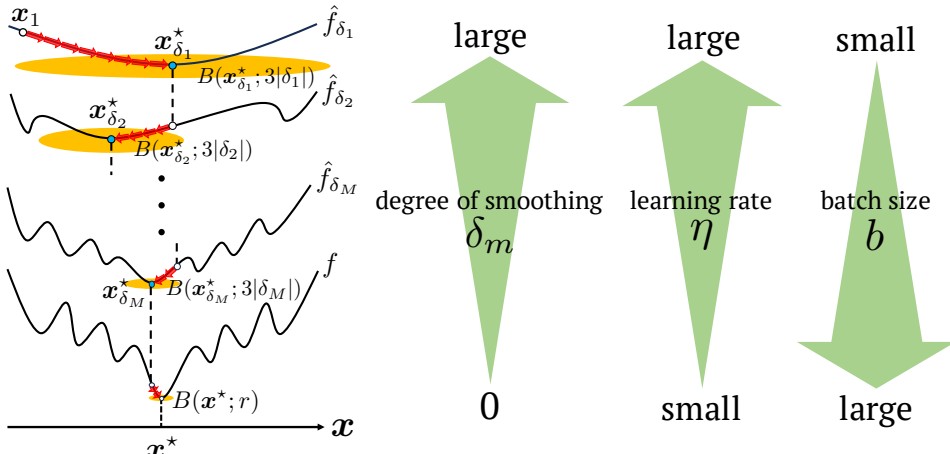

Figure 1: Conceptual diagram of implicit graduated optimization for $\sigma_m$-nice function.

example, as is well known, training with a large batch size leads to poor generalization performance, as evidenced by the fact that several prior studies (Hoffer et al., 2017; Goyal et al., 2017; You et al., 2020) provided techniques that do not impair generalization performance even with large batch sizes. This is because, if we use a large batch size, the degree of smoothing $\delta = \frac{\eta C}{\sqrt{b}}$ becomes smaller and the original nonconvex function is not smoothed enough, so the sharp local minima do not disappear and the optimizer is more likely to fall into one. (Keskar et al., 2017) showed this experimentally, and our results provide theoretical support for it.

**2. Relationship between degree of smoothing, sharpness, and generalizability (Section 4).** To support our theory that simply using SGD for optimization smoothes the objective function and that the degree of smoothing is determined by $\delta = \eta C / \sqrt{b}$, we experimentally confirmed the relationship between the sharpness of the function around the approximate solution to which the optimizer converges and the degree of smoothing. We showed that the degree of smoothing is clearly able to express the smoothness/sharpness of the function as well as the well-studied "sharpness" indicator (Figure 2 (A)), and that it is more strongly correlated with generalization performance than sharpness (Figure 2 (B) and (C)). Our results follow up on a previous study (Andriushchenko et al., 2023) that found, through extensive experiments, correlations between generalization performance and hyperparameters such as the learning rate, but no correlation between it and sharpness.

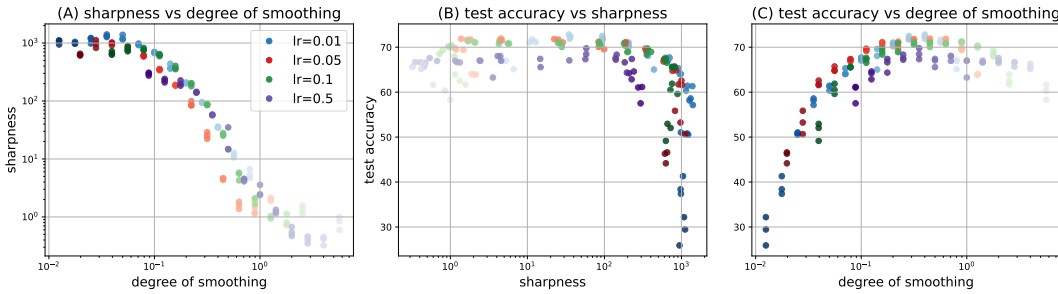

Figure 2: **(A)** Sharpness versus degree of smoothing calculated from learning rate, batch size, and the estimated variance of the stochastic gradient. **(B)** Test accuracy after 200 epochs ResNet18 training on the CIFAR100 dataset versus sharpness. **(C)** Test accuracy versus degree of smoothing. The color shading in the scatter plots represents the batch size: the larger the batch size, the darker the color of the plotted points. "lr" means learning rate.

**3. Implicit Graduated Optimization (Section 5).** Since the degree of smoothing of the objective function by stochastic noise in SGD is determined by $\delta = \frac{\eta C}{\sqrt{b}}$, it should be possible to construct an implicit graduated optimization algorithm by decreasing the learning rate and/or increasing the batch size during training. Based on this theoretical intuition, we propose a new implicit graduated optimization algorithm and $\sigma_m$-nice function which slightly extend $\sigma$-nice function. We also show

that the algorithm for the $\sigma_m$-nice function converges to an $\epsilon$-neighborhood of the global optimal solution in $\mathcal{O}\left(1/\epsilon^2\right)$ rounds. In Section 5.2, we show experimentally that our implicit graduated algorithm outperforms SGD using a constant learning rate and constant batch size. We also find that methods which increase the batch size outperform those which decrease the learning rate when the decay rate of the degree of smoothing is set at $1/\sqrt{2}$.

## 2 PRELIMINARIES

Let $\mathbb{N}$ be the set of non-negative integers. For $m \in \mathbb{N} \setminus \{0\}$, define $[m] := \{1, 2, \ldots, m\}$. Let $\mathbb{R}^d$ be a $d$-dimensional Euclidean space with inner product $\langle \cdot, \cdot \rangle$, which induces the norm $\|\cdot\|$. $I_d$ denotes a $d \times d$ identity matrix. $B(\boldsymbol{y}; r)$ is the Euclidean closed ball of radius $r$ centered at $\boldsymbol{y}$, i.e., $B(\boldsymbol{y}; r) := \left\{\boldsymbol{x} \in \mathbb{R}^d : \|\boldsymbol{x} - \boldsymbol{y}\| \leq r\right\}$. Let $\mathcal{N}(\boldsymbol{\mu}; \Sigma)$ be a $d$-dimensional Gaussian distribution with mean $\boldsymbol{\mu} \in \mathbb{R}^d$ and variance $\Sigma \in \mathbb{R}^{d \times d}$. The DNN is parameterized by a vector $\boldsymbol{x} \in \mathbb{R}^d$, which is optimized by minimizing the empirical loss function $f(\boldsymbol{x}) := \frac{1}{n} \sum_{i \in [n]} f_i(\boldsymbol{x})$, where $f_i(\boldsymbol{x})$ is the loss function for $\boldsymbol{x} \in \mathbb{R}^d$ and the $i$-th training data $\boldsymbol{z}_i$ ($i \in [n]$). Let $\xi$ be a random variable that does not depend on $\boldsymbol{x} \in \mathbb{R}^d$, and let $\mathbb{E}_\xi[X]$ denote the expectation with respect to $\xi$ of a random variable $X$. $\xi_{t,i}$ is a random variable generated from the $i$-th sampling at time $t$, and $\boldsymbol{\xi}_t := (\xi_{t,1}, \xi_{t,2}, \ldots, \xi_{t,b})$ is independent of $(\boldsymbol{x}_k)_{k=0}^t \subset \mathbb{R}$, where $b$ ($\leq n$) is the batch size. The independence of $\boldsymbol{\xi}_0, \boldsymbol{\xi}_1, \ldots$ allows us to define the total expectation $\mathbb{E}$ as $\mathbb{E} = \mathbb{E}_{\boldsymbol{\xi}_0} \mathbb{E}_{\boldsymbol{\xi}_1} \cdots \mathbb{E}_{\boldsymbol{\xi}_t}$. Let $\mathsf{G}_{\boldsymbol{\xi}_t}(\boldsymbol{x})$ be the stochastic gradient of $f(\cdot)$ at $\boldsymbol{x} \in \mathbb{R}^d$. The mini-batch $\mathcal{S}_t$ consists of $b$ samples at time $t$, and the mini-batch stochastic gradient of $f(\boldsymbol{x}_t)$ for $\mathcal{S}_t$ is defined as $\nabla f_{\mathcal{S}_t}(\boldsymbol{x}_t) := \frac{1}{b} \sum_{i \in [b]} \mathsf{G}_{\xi_{t,i}}(\boldsymbol{x}_t)$.

**Definition 2.1** (Smoothed function). *Given a function $f: \mathbb{R}^d \to \mathbb{R}$, define $\hat{f}_\delta: \mathbb{R}^d \to \mathbb{R}$ to be the function obtained by smoothing $f$ as $\hat{f}_\delta(\boldsymbol{x}) := \mathbb{E}_{\boldsymbol{u} \sim \mathcal{L}}[f(\boldsymbol{x} - \delta \boldsymbol{u})]$, where $\delta > 0$ represents the degree of smoothing and $\boldsymbol{u}$ is a random variable from a any light-tailed distribution $\mathcal{L}$ with $\mathbb{E}_{\boldsymbol{u} \sim \mathcal{L}}[\|\boldsymbol{u}\|] \leq 1$. Also, $\boldsymbol{x}^\star := \operatorname*{argmin}_{\boldsymbol{x} \in \mathbb{R}^d} f(\boldsymbol{x})$ and $\boldsymbol{x}_\delta^\star := \operatorname*{argmin}_{\boldsymbol{x} \in \mathbb{R}^d} \hat{f}_\delta(\boldsymbol{x})$.*

Note that, in the definition of the smoothed function $\hat{f}_{\delta_m}$, (Hazan et al., 2016) defined that the random variable $\boldsymbol{u}$ follows a uniform distribution from the unit Euclidean ball. In contrast, from experimental results in Section H, we define the random variable $\boldsymbol{u}$ to follow any light-tailed distribution. Since the uniform distribution is a light-tailed distribution (see Section H.1), our Definition 2.1 contains Definition 4.1 of (Hazan et al., 2016) and does not conflict with it. The graduated optimization algorithm uses several smoothed functions with different noise levels. There are a total of $M$ noise levels $(\delta_m)_{m \in [M]}$ and smoothed functions $(\hat{f}_{\delta_m})_{m \in [M]}$ in this paper. The largest noise level is $\delta_1$ and the smallest is $\delta_M$ (see also Figure 1). For all $m \in [M]$, $(\hat{\boldsymbol{x}}_t^{(m)})_{t \in \mathbb{N}}$ is the sequence generated by an optimizer to minimize $\hat{f}_{\delta_m}$. Here, this paper refers to the graduated optimization approach with explicit smoothing operations (Definition 2.1) as "explicit graduated optimization" and to the graduated optimization approach with implicit smoothing operations as "implicit graduated optimization". All previous studies (see Section 1.1) have considered explicit graduated optimization, and we consider implicit graduated optimization for the first time.

We make the following assumptions:

**Assumption 2.1.** *(A1) $f: \mathbb{R}^d \to \mathbb{R}$ is continuously differentiable and $L_g$-smooth, i.e., for all $\boldsymbol{x}, \boldsymbol{y} \in \mathbb{R}^d$, $\|\nabla f(\boldsymbol{x}) - \nabla f(\boldsymbol{y})\| \leq L_g \|\boldsymbol{x} - \boldsymbol{y}\|$. (A2) $f: \mathbb{R}^d \to \mathbb{R}$ is an $L_f$-Lipschitz function, i.e., for all $\boldsymbol{x}, \boldsymbol{y} \in \mathbb{R}^d$, $|f(\boldsymbol{x}) - f(\boldsymbol{y})| \leq L_f \|\boldsymbol{x} - \boldsymbol{y}\|$. (A3) Let $(\boldsymbol{x}_t)_{t \in \mathbb{N}} \subset \mathbb{R}^d$ be the sequence generated by SGD. (i) For each iteration $t$, $\mathbb{E}_{\xi_t}[\mathsf{G}_{\xi_t}(\boldsymbol{x}_t)] = \nabla f(\boldsymbol{x}_t)$. (ii) There exists a nonnegative constant $C^2$ such that $\mathbb{E}_{\xi_t}[\|\mathsf{G}_{\xi_t}(\boldsymbol{x}_t) - \nabla f(\boldsymbol{x}_t)\|^2] \leq C^2$. (A4) For each iteration $t$, SGD samples a mini-batch $\mathcal{S}_t \subset \mathcal{S}$ and estimates the full gradient $\nabla f$ as $\nabla f_{\mathcal{S}_t}(\boldsymbol{x}_t) := \frac{1}{b} \sum_{i \in [b]} \mathsf{G}_{\xi_{t,i}}(\boldsymbol{x}_t) = \frac{1}{b} \sum_{\{i: \boldsymbol{z}_i \in \mathcal{S}_t\}} \nabla f_i(\boldsymbol{x}_t)$.*

The proof of Lemmas 2.1 and 2.2 can be found in Appendix C.

**Lemma 2.1.** *Suppose that (A3)(ii) and (A4) hold for all $t \in \mathbb{N}$; then, $\mathbb{E}_{\xi_t}[\|\nabla f_{\mathcal{S}_t}(\boldsymbol{x}_t) - \nabla f(\boldsymbol{x}_t)\|^2] \leq \frac{C^2}{b}$.*

**Lemma 2.2.** *Let $\hat{f}_\delta$ be the smoothed version of $f$; then, for all $\boldsymbol{x} \in \mathbb{R}^d$, $\left|\hat{f}_\delta(\boldsymbol{x}) - f(\boldsymbol{x})\right| \leq \mathbb{E}_{\boldsymbol{u}}[\|\boldsymbol{u}\|] \delta L_f$.*

The graduated optimization algorithm is a method in which the degree of smoothing $\delta$ is gradually decreased. Let us consider the case where the degree of smoothing $\delta$ is constant throughout the training. Here, a larger degree of smoothing should be necessary to make many local optimal solutions of the objective function $f$ disappear and lead the optimizer to the global optimal solution. On the other hand, Lemma 2.2 implies that the larger the degree of smoothing is, the further away the smoothed function will be from the original function. Therefore, there should be an optimal value for the degree of smoothing that balances the tradeoffs, because if the degree of smoothing is too large, the original function is too damaged and thus cannot be optimized properly, and if it is too small, the function is not smoothed enough and the optimizer falls into a local optimal solution. This knowledge is useful because the degree of smoothing due to stochastic noise in SGD is determined by the learning rate and batch size (see Section 3), so when a constant learning rate and constant batch size are used, the degree of smoothing is constant throughout the training (see Section 4).

## 3 SGD's SMOOTHING PROPERTY

This section discusses the smoothing effect of using stochastic gradients. From Lemma 2.1, we have

$$\mathbb{E}_{\xi_t}\left[\|\boldsymbol{\omega}_t\|\right] \leq \frac{C}{\sqrt{b}},$$

due to $\boldsymbol{\omega}_t := \nabla f_{\mathcal{S}_t}(\boldsymbol{x}_t) - \nabla f(\boldsymbol{x}_t)$. The $\boldsymbol{\omega}_t$ for which this equation is satisfied can be expressed as $\boldsymbol{\omega}_t = \frac{C}{\sqrt{b}}\boldsymbol{u}_t$, where $\mathbb{E}_{\xi_t}\left[\|\boldsymbol{u}_t\|\right] \leq 1$. Here, we assume that $\boldsymbol{\omega}_t$ in image classification tasks with CNN-based models follows a light-tailed distribution in accordance with experimental observations in several previous studies (Zhang et al., 2020; Kunstner et al., 2023) and our experimental results (see Section H.2). Therefore, $\boldsymbol{\omega}_t \sim \hat{\mathcal{L}}$ and thereby $\boldsymbol{u}_t \sim \mathcal{L}$, where $\hat{\mathcal{L}}$ and $\mathcal{L}$ are light-tailed distributions and $\mathcal{L}$ is a scaled version of $\hat{\mathcal{L}}$. Then, using Definition 2.1, we further transform equation (1) as follows:

$$\mathbb{E}_{\boldsymbol{\omega}_t}\left[\boldsymbol{y}_{t+1}\right] = \mathbb{E}_{\boldsymbol{\omega}_t}\left[\boldsymbol{y}_t\right] - \eta\nabla\mathbb{E}_{\boldsymbol{\omega}_t}\left[f(\boldsymbol{y}_t - \eta\boldsymbol{\omega}_t)\right]$$

$$= \mathbb{E}_{\boldsymbol{\omega}_t}\left[\boldsymbol{y}_t\right] - \eta\nabla\mathbb{E}_{\boldsymbol{u}_t\sim\mathcal{L}}\left[f\left(\boldsymbol{y}_t - \frac{\eta C}{\sqrt{b}}\boldsymbol{u}_t\right)\right]$$

$$= \mathbb{E}_{\boldsymbol{\omega}_t}\left[\boldsymbol{y}_t\right] - \eta\nabla\hat{f}_{\frac{\eta C}{\sqrt{b}}}(\boldsymbol{y}_t). \tag{2}$$

This shows that $\mathbb{E}_{\boldsymbol{\omega}_t}\left[f(\boldsymbol{y}_t - \eta\boldsymbol{\omega}_t)\right]$ is a smoothed version of $f$ with a noise level $\eta C/\sqrt{b}$ and its parameter $\boldsymbol{y}_t$ can be approximately updated by using the gradient descent to minimize $\hat{f}_{\frac{\eta C}{\sqrt{b}}}$. Therefore, we can say that the degree of smoothing $\delta$ by the stochastic noise $\boldsymbol{\omega}_t$ in SGD is determined by the learning rate $\eta$, the batch size $b$, and the variance of the stochastic gradient $C^2$ and that optimizing the function $f$ with SGD and optimizing the smoothed function $\hat{f}_{\frac{\eta C}{\sqrt{b}}}$ with GD are equivalent in the sense of expectation.

There are still more discoveries that can be made from the finding that simply by using SGD for optimization, the objective function is smoothed and the degree of smoothing is determined by $\delta = \eta C/\sqrt{b}$.

**Why the Use of Large Batch Sizes Leads to Solutions Falling into Sharp Local Minima.** It is known that training with large batch sizes leads to a persistent degradation of model generalization performance. In particular, (Keskar et al., 2017) showed experimentally that learning with large batch sizes leads to sharp local minima and worsens generalization performance. According to equation (2), using a large learning rate and/or a small batch size will make the function smoother. Thus, in using a small batch size, the sharp local minima will disappear through extensive smoothing, and SGD can reach a flat local minimum. Conversely, when using a large batch size, the smoothing is weak and the function is close to the original multimodal function, so it is easy for the solution to fall into a sharp local minimum. Thus, we have theoretical support for what (Keskar et al., 2017) showed experimentally, and our experiments have yielded similar results (see Figure 3 (a) and (e)).

**Why Decaying Learning Rates and Increasing Batch Sizes are Superior to Fixed Learning Rates and Batch Sizes.** From equation (2), the use of a decaying learning rate or increasing batch size during training is equivalent to decreasing the noise level of the smoothed function, so using

a decaying learning rate or increasing the batch size is an implicit graduated optimization. Thus, we can say that using a decaying learning rate (Loshchilov & Hutter, 2017; Hundt et al., 2019; You et al., 2019; Lewkowycz, 2021) or increasing batch size (Byrd et al., 2012; Friedlander & Schmidt, 2012; Balles et al., 2017; De et al., 2017; Bottou et al., 2018; Smith et al., 2018) makes sense in terms of avoiding local minima and provides theoretical support for their experimental superiority.

## 4 RELATIONSHIP BETWEEN DEGREE OF SMOOTHING, SHARPNESS, AND GENERALIZABILITY

The smoothness of the function, and in particular the sharpness of the function around the approximate solution to which the optimizer converged, has been well studied because it has been thought to be related to the generalizability of the model. In this section, we reinforce our theory by experimentally observing the relationship between the degree of smoothing and the sharpness of the function.

Several previous studies (Hochreiter & Schmidhuber, 1997; Keskar et al., 2017; Izmailov et al., 2018; Li et al., 2018; Andriushchenko et al., 2023) have addressed the relationship between the sharpness of the function around the approximate solution to which the optimizer converges and the generalization performance of the model. In particular, the hypothesis that flat local solutions have better generalizability than sharp local solutions is at the core of a series of discussions, and several previous studies (Keskar et al., 2017; Liang et al., 2019; Tsuzuku et al., 2020; Petzka et al., 2021; Kwon et al., 2021) have developed measures of sharpness to confirm this. In this paper, we use "adaptive sharpness" (Kwon et al., 2021; Andriushchenko et al., 2023) as a measure of the sharpness of the function that is invariant to network reparametrization, highly correlated with generalization, and generalizes several existing sharpness definitions. In accordance with (Andriushchenko et al., 2023), let $\mathcal{S}$ be a set of training data; for arbitrary model weights $\boldsymbol{w} \in \mathbb{R}^d$, the worst-case adaptive sharpness with radius $\rho \in \mathbb{R}$ and with respect to a vector $\boldsymbol{c} \in \mathbb{R}^d$ is defined as

$$S_{\max}^{\rho}(\boldsymbol{w}, \boldsymbol{c}) := \mathbb{E}_{\mathcal{S}} \left[ \max_{\|\boldsymbol{\delta} \odot \boldsymbol{c}^{-1}\|_p \leq \rho} f(\boldsymbol{w} + \boldsymbol{\delta}) - f(\boldsymbol{w}) \right],$$

where $\odot/^{-1}$ denotes elementwise multiplication/inversion. Thus, the larger the sharpness value is, the sharper the function around the model weight $\boldsymbol{w}$ becomes, with a smaller sharpness leading to higher generalizability.

We trained ResNet18 (He et al., 2016) with the learning rate $\eta \in \{0.01, 0.05, 0.1, 0.1\}$ and batch size $b \in \{2^1, \ldots, 2^{13}\}$ for 200 epochs on the CIFAR100 dataset (Krizhevsky, 2009) and then measured the worst-case $l_\infty$ adaptive sharpness of the obtained approximate solution with radius $\rho = 0.0002$ and $\boldsymbol{c} = (1, 1, \ldots, 1)^{\top} \in \mathbb{R}^d$. Our implementation was based on (Andriushchenko et al., 2023) and the code used is available on our anonymous Github. Figure 3 plots the relationship between measured sharpness and the batch size $b$ and the learning rate $\eta$ used for training as well as the degree of smoothing $\delta$ calculated from them. Figure 3 also plots the relationship between test accuracy, sharpness, and degree of smoothing. Three experiments were conducted per combination of learning rate and batch size, with a total of 156 data plots. The variance of the stochastic gradient $C^2$ included in the degree of smoothing $\delta = \eta C/\sqrt{b}$ used values estimated from theory and experiment (see Appendix B for details).

Figure 3 (a) shows that the larger the batch size is, the larger the sharpness value becomes, whereas (b) shows that the larger the learning rate is, the smaller the sharpness becomes, and (c) shows a greater the degree of smoothing for a smaller sharpness. These experimental results guarantee the our theoretical result that the degree of smoothing $\delta$ is proportional to the learning rate $\eta$ and inversely proportional to the batch size $b$, and they reinforce our theory that the quantity $\eta C/\sqrt{b}$ is the degree of smoothing of the function. Figure 3 (d) also shows that there is no clear correlation between the generalization performance of the model and the sharpness around the approximate solution. This result is also consistent with previous study (Andriushchenko et al., 2023). On the other hand, Figure 3 (e) shows an excellent correlation between generalization performance and the degree of smoothing; generalization performance is clearly a concave function with respect to the degree of smoothing. Thus, a degree of smoothing that is neither too large nor too small leads to high generalization performance. This experimental observation can be supported theoretically (see Lemma 2.2). That is, if the degree of smoothing is a constant throughout the training, then there

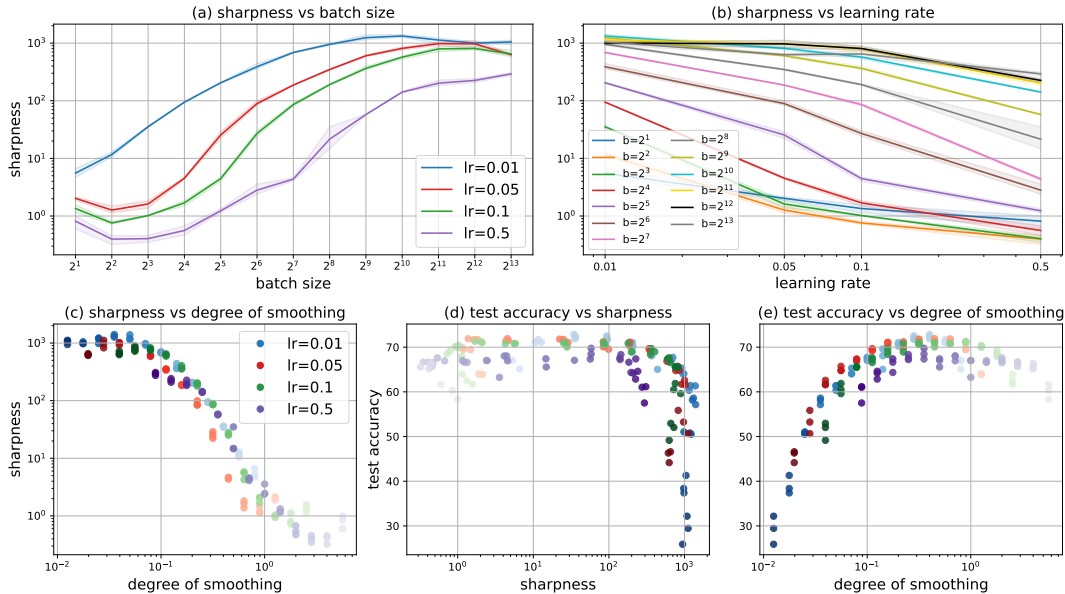

Figure 3: **(a)** Sharpness around the approximate solution after 200 epochs of ResNet18 training on the CIFAR100 dataset versus batch size used. **(b)** Sharpness versus learning rate used. **(c)** Sharpness versus degree of smoothing calculated from learning rate, batch size and estimated variance of the stochastic gradient. **(d)** Test accuracy after 200 epochs training versus sharpness. **(e)** Test accuracy versus degree of smoothing. The solid line represents the mean value, and the shaded area represents the maximum and minimum over three runs. The color shade in the scatter plots represents the batch size; the larger the batch size, the darker the color of the plotted points. "lr" means learning rate. The experimental results that make up the all graphs are all identical.

should be an optimal value for the loss function value or test accuracy; for the training of ResNet18 on the CIFAR100 dataset, for example, 0.1 to 1 was the desired value (see Figure 3 (e)). For degrees of smoothing smaller than 0.1, the generalization performance is not good because the function is not sufficiently smoothed so that locally optimal solutions with sharp neighborhoods do not disappear, and the optimizer falls into this trap. On the other hand, a degree of smoothing greater than 1 leads to excessive smoothing and smoothed function becomes too far away from the original function to be properly optimized; the generalization performance is not considered excellent. In addition, the optimal combination of learning rate and batch size that practitioners search for by using grid searches or other methods when training models can be said to be a search for the optimal degree of smoothing. If the optimal degree of smoothing can be better understood, the huge computational cost of the search could be reduced.

(Andriushchenko et al., 2023) observed the relationship between sharpness and generalization performance in extensive experiments and found that they were largely uncorrelated, suggesting that the sharpnesss may not be a good indicator of generalization performance and that one should avoid blanket statements like "flatter minima generalize better". Figure 3 (d) and (e) show that there is no correlation between sharpness and generalization performance, as in previous study, while there is a correlation between degree of smoothing and generalization performance. Therefore, we can say that degree of smoothing may be a good indicator to theoretically evaluate generalization performance, and it may be too early to say that "flatter minima generalize better" is invalid.

## 5 IMPLICIT GRADUATED OPTIMIZATION

In this section, we construct an implicit graduated optimization algorithm that varies the learning rate $\eta$ and batch size $b$ so that the degree of smoothing $\delta = \eta C/\sqrt{b}$ by stochastic noise in SGD gradually decreases and then analyze its convergence.

## 5.1 ANALYSIS OF IMPLICIT GRADUATED OPTIMIZATION ALGORITHM

In order to analyze the graduated optimization algorithm, Hazan et al. defined $\sigma$-nice functions (see Definition I.1), a family of nonconvex functions that has favorable conditions for a graduated optimization algorithm to converge to a global optimal solution (Hazan et al., 2016). We define the following function, which is a slight extension of the $\sigma$-nice function. See Section I for details on its extension.

**Definition 5.1** ($\sigma_m$-nice function). *Let $M \in \mathbb{N}$, $m \in [M]$, and $\gamma \in [0.5, 1)$. A function $f \colon \mathbb{R}^d \to \mathbb{R}$ is said to be $\sigma_m$-nice if the following two conditions hold:*

*(i) For all $\delta_m > 0$ and all $\boldsymbol{x}^\star_{\delta_m}$, there exists $\boldsymbol{x}^\star_{\delta_{m+1}}$ such that: $\left\| \boldsymbol{x}^\star_{\delta_m} - \boldsymbol{x}^\star_{\delta_{m+1}} \right\| \le \delta_{m+1} := \gamma \delta_m$.*

*(ii) For all $\delta_m > 0$, the function $\hat{f}_{\delta_m}(\boldsymbol{x})$ over $N(\boldsymbol{x}^\star_{\delta_m}; 3\delta_m)$ is $\sigma_m$-strongly convex.*

The $\sigma_m$-nice property implies that optimizing the smoothed function $\hat{f}_{\delta_m}$ is a good start for optimizing the next smoothed function $\hat{f}_{\delta_{m+1}}$, which has been shown to be sufficient for graduated optimization (Hazan et al., 2016). We will take this route and consider an implicit graduated optimization algorithm for $\sigma_m$-nice functions.

Algorithm 1 embodies the framework of implicit graduated optimization with SGD for $\sigma_m$-nice functions, while Algorithm 2 is used to optimize each smoothed function; it should be GD (see (2)). Note that our implicit graduated optimization (Algorithm 1) is achieved by SGD with decaying learning rate and/or increasing batch size.

---

**Algorithm 1** Implicit Graduated Optimization

**Require:** $\epsilon, \boldsymbol{x}_1 \in B(\boldsymbol{x}^\star_{\delta_1}; 3\delta_1), \eta_1 > 0, b_1 \in [n], \gamma \ge 0.5$
$\quad \delta_1 := \frac{\eta_1 C}{\sqrt{b_1}}, \alpha_0 := \min\left\{ \frac{1}{16 L_f \delta_1}, \frac{1}{\sqrt{2\sigma \delta_1}} \right\}, M := \log_\gamma \alpha_0 \epsilon$
$\quad$ **for** $m = 1$ to $M + 1$ **do**
$\quad\quad$ **if** $m \ne M + 1$ **then**
$\quad\quad\quad \epsilon_m := \sigma_m \delta_m^2 / 2, \ T_m := H_m / \epsilon_m$
$\quad\quad\quad \kappa_m / \sqrt{\lambda_m} = \gamma \ (\kappa_m \in (0, 1], \lambda_m \ge 1)$
$\quad\quad$ **end if**
$\quad\quad \boldsymbol{x}_{m+1} := \mathrm{GD}(T_m, \boldsymbol{x}_m, \hat{f}_{\delta_m}, \eta_m)$
$\quad\quad \eta_{m+1} := \kappa_m \eta_m, b_{m+1} := \lambda_m b_m$
$\quad\quad \delta_{m+1} := \frac{\eta_{m+1} C}{\sqrt{b_{m+1}}}$
$\quad$ **end for**
$\quad$ **return** $\boldsymbol{x}_{M+2}$

---

**Algorithm 2** Gradient Descent

**Require:** $T_m, \hat{\boldsymbol{x}}^{(m)}_1, \hat{f}_{\delta_m}, \eta > 0$
$\quad$ **for** $t = 1$ to $T_m$ **do**
$\quad\quad \hat{\boldsymbol{x}}^{(m)}_{t+1} := \hat{\boldsymbol{x}}^{(m)}_t - \eta \nabla \hat{f}_{\delta_m}(\boldsymbol{x}_t)$
$\quad$ **end for**
$\quad$ **return** $\hat{\boldsymbol{x}}^{(m)}_{T_m + 1}$

---

From the definition of $\sigma_m$-nice function, the smoothed function $\hat{f}_{\delta_m}$ is $\sigma_m$-strongly convex in $B(\boldsymbol{x}^\star_{\delta_m}; 3\delta_m)$. Also, the learning rate used by Algorithm 2 to optimize $\hat{f}_{\delta_m}$ should always be constant. Therefore, let us now consider the convergence of GD with a constant learning rate for a $\sigma_m$-strongly convex function $\hat{f}_{\delta_m}$. The proof of Theorem 5.1 is in Appendix F.1.

**Theorem 5.1** (Convergence analysis of Algorithm 2). *Suppose that $\hat{f}_{\delta_m} \colon \mathbb{R}^d \to \mathbb{R}$ is a $\sigma_m$-strongly convex and $L_g$-smooth and $\eta < \min\left\{ \frac{1}{\sigma_m}, \frac{2}{L_g} \right\}$. Then, the sequence $(\hat{\boldsymbol{x}}^{(m)}_t)_{t \in \mathbb{N}}$ generated by Algorithm 2 satisfies*

$$\min_{t \in [T]} \hat{f}_{\delta_m}\left( \hat{\boldsymbol{x}}^{(m)}_t \right) - \hat{f}_{\delta_m}(\boldsymbol{x}^\star_{\delta_m}) \le \frac{H_m}{T} = \mathcal{O}\left( \frac{1}{T} \right), \tag{3}$$

*where $H_m := \frac{9(1 - \sigma_m \eta)\delta_m^2}{2\eta} + \frac{3 L_f \delta_m}{\eta(2 - L_g \eta)}$ is a nonnegative constant.*

Theorem 5.1 shows that Algorithm 2 can reach an $\epsilon_m$-neighborhood of the optimal solution $\boldsymbol{x}^\star_{\delta_m}$ of $\hat{f}_{\delta_m}$ in approximately $T_m := H_m / \epsilon_m$ iterations. The next proposition is crucial to the success of Algorithm 1 and guarantees the soundness of the $\sigma_m$-nice function (The proof is in Appendix F.2).

**Proposition 5.1.** *Let $f$ be a $\sigma_m$-nice function and $\delta_{m+1} := \gamma \delta_m$. Suppose that $\gamma \in [0.5, 1)$ and $\boldsymbol{x}_1 \in B(\boldsymbol{x}^\star_{\delta_1}; 3\delta_1)$. Then for all $m \in [M]$, $\|\boldsymbol{x}_m - \boldsymbol{x}^\star_{\delta_m}\| < 3\delta_m$.*

$\boldsymbol{x}_m$ is the approximate solution obtained by optimization of the smoothed function $\hat{f}_{\delta_{m-1}}$ with Algorithm 2 and is the initial point of optimization of the next smoothed function $\hat{f}_{\delta_m}$. Therefore, Proposition 5.1 implies that $\gamma \in [0.5, 1)$ must hold for the initial point of optimization of $\hat{f}_{\delta_m}$ to be contained in the strongly convex region of $\hat{f}_{\delta_m}$. Therefore, from Theorem 5.1 and Proposition 5.1, if $f$ is a $\sigma_m$-nice function and $\boldsymbol{x}_1 \in B(\boldsymbol{x}_{\delta_1}^{\star}; 3\delta_1)$ holds, the sequence $(\boldsymbol{x}_m)_{m \in [M]}$ generated by Algorithm 1 never goes outside of the $\sigma_m$-strongly convex region $B(\boldsymbol{x}_{\delta_m}^{\star}; 3\delta_m)$ of each smoothed function $\hat{f}_{\delta_m}$ $(m \in [M])$.

The next theorem guarantees the convergence of Algorithm 1 with the $\sigma_m$-nice function (The proof of Theorem 5.2 is in Appendix F.3).

**Theorem 5.2** (Convergence analysis of Algorithm 1). *Let $\epsilon \in (0, 1)$ and $f \colon \mathbb{R}^d \to \mathbb{R}$ be an $L_f$-Lipschitz $\sigma_m$-nice function. Suppose that we run Algorithm 1; then after $\mathcal{O}\left(1/\epsilon^2\right)$ rounds, the algorithm reaches an $\epsilon$-neighborhood of the global optimal solution $\boldsymbol{x}^{\star}$.*

Note that Theorem 5.2 provides a total complexity including those of Algorithm 1 and Algorithm 2, because Algorithm 1 uses Algorithm 2 at each $m \in [M]$.

### 5.2 NUMERICAL RESULTS

The experimental environment is in Appendix G and the code is available at https://anonymous.4open.science/r/new-sigma-nice.

We compared four types of SGD for image classification: 1. constant learning rate and constant batch size, 2. decaying learning rate and constant batch size, 3. constant learning rate and increasing batch size, 4. decaying learning rate and increasing batch size, in training ResNet34 (He et al., 2016) on the ImageNet dataset (Deng et al., 2009) (Figure 4), ResNet18 on the CIFAR100 dataset (Figure 5 in Appendix G), and WideResNet-28-10 (Zagoruyko & Komodakis, 2016) on the CIFAR100 dataset (Figure 6 in Appendix G). Therefore, methods 2, 3, and 4 are our Algorithm 1. All experiments were run for 200 epochs. In methods 2, 3, and 4, the noise decreased every 40 epochs, with a common decay rate of $1/\sqrt{2}$. That is, every 40 epochs, the learning rate of method 2 was multiplied by $1/\sqrt{2}$, the batch size of method 3 was doubled, and the learning rate and batch size of method 4 were respectively multiplied by $\sqrt{3}/2$ and 1.5. Note that this $1/\sqrt{2}$ decay rate is $\gamma$ in Algorithm 1 and it satisfies the condition in Proposition 5.1. The initial learning rate was 0.1 for all methods, which was determined by performing a grid search among $[0.01, 0.1, 1.0, 10]$. The noise reduction interval was every 40 epochs, which was determined by performing a grid search among $[10, 20, 25, 40, 50, 100]$. A history of the learning rate or batch size for each method is provided in the caption of each figure.

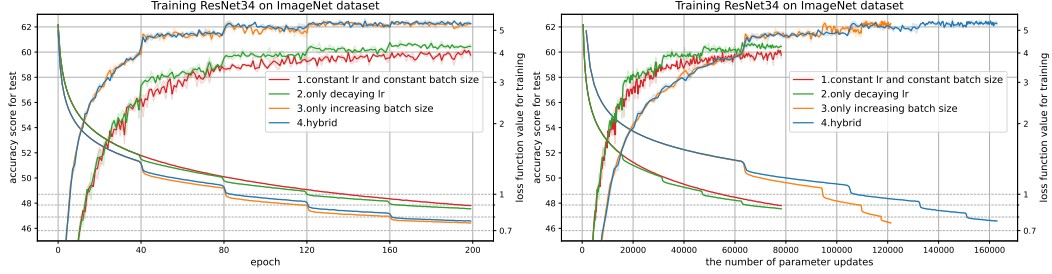

Figure 4: Accuracy score for the testing and loss function value for training versus the number of epochs (**left**) and the number of parameter updates (**right**) in training ResNet34 on the ImageNet dataset. The solid line represents the mean value, and the shaded area represents the maximum and minimum over three runs. In method 1, the learning rate and batch size were fixed at 0.1 and 256, respectively. In method 2, the learning rate was decreased every 40 epochs as $\left[0.1, \frac{1}{10\sqrt{2}}, 0.05, \frac{1}{20\sqrt{2}}, 0.025\right]$ and the batch size was fixed at 256. In method 3, the learning rate was fixed at 0.1, and the batch size was increased as $[32, 64, 128, 256, 512]$. In method 4, the learning rate was decreased as $\left[0.1, \frac{\sqrt{3}}{20}, 0.075, \frac{3\sqrt{3}}{80}, 0.05625\right]$ and the batch size was increased as $[32, 48, 72, 108, 162]$.

For methods 2, 3, and 4, the decay rates are all $1/\sqrt{2}$, and the decay intervals are all 40 epochs, so throughout the training, the three methods should theoretically be optimizing the exact same five smoothed functions in sequence. Nevertheless, the local solutions reached by each of the three methods are not exactly the same. All results indicate that method 3 is superior to method 2 and that method 4 is superior to method 3 in both test accuracy and training loss function values. This difference can be attributed to the different learning rates used to optimize each smoothing function. Among methods 2, 3, and 4, method 3, which does not decay the learning rate, maintains the highest learning rate 0.1, followed by method 4 and method 2. In all graphs, the loss function values are always small in that order; i.e., the larger the learning rate is, the lower loss function values become. Therefore, we can say that the noise level $\delta$, expressed as $\frac{\eta C}{\sqrt{b}}$, needs to be reduced, while the learning rate $\eta$ needs to remain as large as possible. Alternatively, if the learning rate is small, then a large number of iterations are required. Thus, for the same rate of change and the same number of epochs, an increasing batch size is superior to a decreasing learning rate because it can maintain a large learning rate and can be made to iterate a lot when the batch size is small.

Theoretically, the noise level $\delta_m$ should gradually decrease and become zero at the end, so in our algorithm 1, the learning rate $\eta_m$ should be zero at the end or the batch size $b_m$ should match the number of data sets at the end. However, if the learning rate is 0, training cannot proceed, and if the batch size is close to a full batch, it is not feasible from a computational point of view. For this reason, the experiments described in this paper are not fully graduated optimizations; i.e., full global optimization is not achieved. In fact, the last batch size used by method 2 is around 128 to 512, which is far from a full batch. Therefore, the solution reached in this experiment is the optimal one for a function that has been smoothed to some extent, and to achieve a global optimization of the DNN, it is necessary to increase only the batch size to eventually reach a full batch, or increase the number of iterations accordingly while increasing the batch size and decaying the learning rate.

## 6 CONCLUSION

We proved that SGD with a mini-batch stochastic gradient has the effect of smoothing the function, and the degree of smoothing is greater with larger learning rates and smaller batch sizes. This shows theoretically that smoothing with large batch sizes is makes it easy to fall into sharp local minima and that using a decaying learning rate and/or increasing batch size is implicitly graduated optimization, which makes sense in the sense that it avoids local optimal solutions. Based on these findings, we proposed a new graduated optimization algorithm that uses a decaying learning rate and increasing batch size and analyzed it. We conducted experiments whose results showed the superiority of our recommended framework for image classification tasks on CIFAR100 and ImageNet. In addition, we observed that the degree of smoothing of the function due to stochastic noise in SGD can express the degree of smoothness of the function as well as sharpness does, and that the degree of smoothing is a good indicator of the generalization performance of the model.

## 7 PREVIOUS STUDIES AND OUR NOVELTY

As shown in equation (1), Kleinberg et al. suggested that stochastic noise in SGD smoothes the objective function (Kleinberg et al., 2018), but the degree to which it did so was not analyzed. We theoretically derived the degree of smoothing and experimentally observed the relationship with sharpness to ensure its correctness, which is a novel and valuable result.

We have shown that the degree of smoothing by stochastic noise in SGD is determined by $\delta = \eta C/\sqrt{b}$. This result may remind readers of previous studies (Goyal et al., 2017; Smith et al., 2018; Xie et al., 2021) that investigated the dynamics of SGD and demonstrated how the ratio $\eta/b$ affects the training dynamics. We should emphasize that our $\eta/\sqrt{b}$ is derived from a completely different point of view, and in particular, the finding that the quantity $\eta C/\sqrt{b}$ contributes to the smoothing of the objective function can only be obtained from our theory.

We use the $\sigma_m$-nice function which slightly extend $\sigma$-nice function proposed by (Hazan et al., 2016) to analyze the implicit graduated optimization algorithm. Technically, the difference between our work and theirs is that they optimize each smoothed function with projected gradient descent with a decaying learning rate, whereas we optimize with gradient descent with a constant learning rate given our theoretical motivation (see (2)). Furthermore, all previous studies (see Section 1.1), in-

cluding the work of Hazan et al. consider explicit graduated optimization. Our implicit graduated optimization with stochastic noise in the optimizer is a completely new idea, and this is the first paper to apply the graduated optimization algorithm to the training of deep learning models on a modern dataset such as ImageNet.

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

## A    DERIVATION OF EQUATION (1)

Let $\boldsymbol{y}_t$ be the parameter updated by gradient descent (GD) and $\boldsymbol{x}_{t+1}$ be the parameter updated by SGD at time $t$, i.e.,

$$\boldsymbol{y}_t := \boldsymbol{x}_t - \eta \nabla f(\boldsymbol{x}_t),$$
$$\boldsymbol{x}_{t+1} := \boldsymbol{x}_t - \eta \nabla f_{\mathcal{S}_t}(\boldsymbol{x}_t)$$
$$= \boldsymbol{x}_t - \eta(\nabla f(\boldsymbol{x}_t) + \boldsymbol{\omega}_t).$$

Then, we have

$$\boldsymbol{x}_{t+1} := \boldsymbol{x}_t - \eta \nabla f_{\mathcal{S}_t}(\boldsymbol{x}_t)$$
$$= (\boldsymbol{y}_t + \eta \nabla f(\boldsymbol{x}_t)) - \eta \nabla f_{\mathcal{S}_t}(\boldsymbol{x}_t)$$
$$= \boldsymbol{y}_t - \eta \boldsymbol{\omega}_t, \tag{4}$$

from $\boldsymbol{\omega}_t := \nabla f_{\mathcal{S}_t}(\boldsymbol{x}_t) - \nabla f(\boldsymbol{x}_t)$. Hence,

$$\boldsymbol{y}_{t+1} = \boldsymbol{x}_{t+1} - \eta \nabla f(\boldsymbol{x}_{t+1})$$
$$= \boldsymbol{y}_t - \eta \boldsymbol{\omega}_t - \eta \nabla f(\boldsymbol{y}_t - \eta \boldsymbol{\omega}_t).$$

By taking the expectation with respect to $\boldsymbol{\omega}_t$ on both sides, we have, from $\mathbb{E}_{\boldsymbol{\omega}_t}[\boldsymbol{\omega}_t] = \boldsymbol{0}$,

$$\mathbb{E}_{\boldsymbol{\omega}_t}[\boldsymbol{y}_{t+1}] = \mathbb{E}_{\boldsymbol{\omega}_t}[\boldsymbol{y}_t] - \eta \nabla \mathbb{E}_{\boldsymbol{\omega}_t}[f(\boldsymbol{y}_t - \eta \boldsymbol{\omega}_t)],$$

where we have used $\mathbb{E}_{\boldsymbol{\omega}_t}[\nabla f(\boldsymbol{y}_t - \eta \boldsymbol{\omega}_t)] = \nabla \mathbb{E}_{\boldsymbol{\omega}_t}[f(\boldsymbol{y}_t - \eta \boldsymbol{\omega}_t)]$, which holds for a Lipschitz-continuous and differentiable $f$ (Shapiro et al., 2009, Theorem 7.49). In addition, from (4) and $\mathbb{E}_{\boldsymbol{\omega}_t}[\boldsymbol{\omega}_t] = \boldsymbol{0}$, we obtain

$$\mathbb{E}_{\boldsymbol{\omega}_t}[\boldsymbol{x}_{t+1}] = \boldsymbol{y}_t.$$

Therefore, on average, the parameter $\boldsymbol{x}_{t+1}$ of the function $f$ arrived at by SGD coincides with the parameter $\boldsymbol{y}_t$ of the smoothed function $\hat{f}(\boldsymbol{y}_t) := \mathbb{E}_{\boldsymbol{\omega}_t}[f(\boldsymbol{y}_t - \eta \boldsymbol{\omega}_t)]$ arrived at by GD.

## B    ESTIMATION OF VARIANCE OF STOCHASTIC GRADIENT

In Section 4, we need to estimate the variance $C^2$ of the stochastic gradient in order to plot the degree of smoothing $\delta = \eta C/\sqrt{b}$. In general, this is difficult to measure, but several previous studies (Imaizumi & Iiduka, 2024; Sato & Iiduka, 2024) have provided the following estimating formula. For some $\epsilon > 0$, when training until $\frac{1}{T}\sum_{k=1}^{T} \mathbb{E}\left[\|\nabla f(\boldsymbol{x}_k)\|^2\right] \leq \epsilon^2$, the variance of the stochastic gradient can be estimated as

$$C^2 < \frac{b^\star \epsilon^2}{\eta},$$

where $b^\star$ is the batch size that minimizes the amount of computation required for training and $\eta$ is learning rate used in training. We determined the stopping condition $\epsilon$ for each learning rate, measured the batch size that minimized the computational complexity required for the gradient norm of the preceding $t$ steps at time $t$ to average less than $\epsilon$ in training ResNet18 on the CIFAR100 dataset, and estimated the variance of the stochastic gradient by using an estimation formula (see Table 1). Table 2 shows the results of a similar experiment for the training WideResNet(WRN)-28-10 on the CIFAR100 dataset.

Table 1: Learning rate $\eta$ and threshold $\epsilon$ used for training, measured optimal batch size $b^\star$ and estimated variance of the stochastic gradient $C^2$ in training ResNet18 on the CIFAR100 dataset.

| $\eta$ | $\epsilon$ | $b^\star$ | $C^2$ |
|------|------|------|------|
| 0.01 | 1.0 | $2^7$ | 12800 |
| 0.05 | 0.5 | $2^9$ | 1280 |
| 0.1 | 0.5 | $2^{10}$ | 1280 |
| 0.5 | 0.5 | $2^{10}$ | 256 |

Table 2: Learning rate $\eta$ and threshold $\epsilon$ used for training, measured optimal batch size $b^\star$ and estimated variance of the stochastic gradient $C^2$ in training WRN-28-10 on the CIFAR100 dataset.

| $\eta$ | $\epsilon$ | $b^\star$ | $C^2$ |
|------|------|------|------|
| 0.01 | 1.0 | $2^2$ | 400 |
| 0.05 | 0.5 | $2^2$ | 20 |
| 0.1 | 0.5 | $2^2$ | 10 |
| 0.5 | 0.5 | $2^2$ | 2 |

## C  PROOFS OF THE LEMMAS IN SECTION 2

### C.1  PROOF OF LEMMA 2.1

*Proof.* (A3)(ii) and (A4) guarantee that

$$\mathbb{E}_{\xi_t}\left[\|\nabla f_{\mathcal{S}_t}(\boldsymbol{x}_t) - \nabla f(\boldsymbol{x}_t)\|^2\right] = \mathbb{E}_{\xi_t}\left[\left\|\frac{1}{b}\sum_{i=1}^{b}\mathsf{G}_{\xi_{t,i}}(\boldsymbol{x}_t) - \nabla f(\boldsymbol{x}_t)\right\|^2\right]$$

$$= \mathbb{E}_{\xi_t}\left[\left\|\frac{1}{b}\sum_{i=1}^{b}\mathsf{G}_{\xi_{t,i}}(\boldsymbol{x}_t) - \frac{1}{b}\sum_{i=1}^{b}\nabla f(\boldsymbol{x}_t)\right\|^2\right]$$

$$= \mathbb{E}_{\xi_t}\left[\left\|\frac{1}{b}\sum_{i=1}^{b}\left(\mathsf{G}_{\xi_{t,i}}(\boldsymbol{x}_t) - \nabla f(\boldsymbol{x}_t)\right)\right\|^2\right]$$

$$= \frac{1}{b^2}\mathbb{E}_{\xi_t}\left[\left\|\sum_{i=1}^{b}\left(\mathsf{G}_{\xi_{t,i}}(\boldsymbol{x}_t) - \nabla f(\boldsymbol{x}_t)\right)\right\|^2\right]$$

$$= \frac{1}{b^2}\mathbb{E}_{\xi_t}\left[\sum_{i=1}^{b}\left\|\mathsf{G}_{\xi_{t,i}}(\boldsymbol{x}_t) - \nabla f(\boldsymbol{x}_t)\right\|^2\right]$$

$$\leq \frac{C^2}{b}.$$

This completes the proof. □

### C.2  PROOF OF LEMMA 2.2

*Proof.* From Definition 2.1 and (A2), we have, for all $\boldsymbol{x}, \boldsymbol{y} \in \mathbb{R}^d$,

$$\left|\hat{f}_\delta(\boldsymbol{x}) - f(\boldsymbol{x})\right| = |\mathbb{E}_{\boldsymbol{u}}\left[f(\boldsymbol{x} - \delta\boldsymbol{u})\right] - f(\boldsymbol{x})|$$

$$= |\mathbb{E}_{\boldsymbol{u}}\left[f(\boldsymbol{x} - \delta\boldsymbol{u}) - f(\boldsymbol{x})\right]|$$

$$\leq \mathbb{E}_{\boldsymbol{u}}\left[|f(\boldsymbol{x} - \delta\boldsymbol{u}) - f(\boldsymbol{x})|\right]$$

$$\leq \mathbb{E}_{\boldsymbol{u}}\left[L_f\|(\boldsymbol{x} - \delta\boldsymbol{u}) - \boldsymbol{x}\|\right]$$

$$= \delta L_f\mathbb{E}_{\boldsymbol{u}}\left[\|\boldsymbol{u}\|\right]$$

$$\leq \delta L_f.$$

This completes the proof. □

## D  LEMMAS ON SMOOTHED FUNCTION

The following Lemmas concern the properties of smoothed functions $\hat{f}_\delta$.

**Lemma D.1.** *Suppose that (A1) holds; then, $\hat{f}_\delta$ defined by Definition 2.1 is also $L_g$-smooth; i.e., for all $\boldsymbol{x}, \boldsymbol{y} \in \mathbb{R}^d$,*

$$\left\| \nabla \hat{f}_\delta(\boldsymbol{x}) - \nabla \hat{f}_\delta(\boldsymbol{y}) \right\| \le L_g \|\boldsymbol{x} - \boldsymbol{y}\|.$$

*Proof.* From Definition 2.1 and (A1), we have, for all $\boldsymbol{x}, \boldsymbol{y} \in \mathbb{R}^d$,

$$
\begin{aligned}
\left\| \nabla \hat{f}_\delta(\boldsymbol{x}) - \nabla \hat{f}_\delta(\boldsymbol{y}) \right\| &= \|\nabla \mathbb{E}_{\boldsymbol{u}}\left[f(\boldsymbol{x} - \delta\boldsymbol{u})\right] - \nabla \mathbb{E}_{\boldsymbol{u}}\left[f(\boldsymbol{y} - \delta\boldsymbol{u})\right]\| \\
&= \|\mathbb{E}_{\boldsymbol{u}}\left[\nabla f(\boldsymbol{x} - \delta\boldsymbol{u})\right] - \mathbb{E}_{\boldsymbol{u}}\left[\nabla f(\boldsymbol{y} - \delta\boldsymbol{u})\right]\| \\
&= \|\mathbb{E}_{\boldsymbol{u}}\left[\nabla f(\boldsymbol{x} - \delta\boldsymbol{u}) - \nabla f(\boldsymbol{y} - \delta\boldsymbol{u})\right]\| \\
&\le \mathbb{E}_{\boldsymbol{u}}\left[\|\nabla f(\boldsymbol{x} - \delta\boldsymbol{u}) - \nabla f(\boldsymbol{y} - \delta\boldsymbol{u})\|\right] \\
&\le \mathbb{E}_{\boldsymbol{u}}\left[L_g \|(\boldsymbol{x} - \delta\boldsymbol{u}) - (\boldsymbol{y} - \delta\boldsymbol{u})\|\right] \\
&= \mathbb{E}_{\boldsymbol{u}}\left[L_g \|\boldsymbol{x} - \boldsymbol{y}\|\right] \\
&= L_g \|\boldsymbol{x} - \boldsymbol{y}\|.
\end{aligned}
$$

This completes the proof. $\qquad\square$

**Lemma D.2.** *Suppose that (A2) holds; then $\hat{f}_\delta$ is also an $L_f$-Lipschitz function; i.e., for all $\boldsymbol{x}, \boldsymbol{y} \in \mathbb{R}^d$,*

$$\left| \hat{f}_\delta(\boldsymbol{x}) - \hat{f}_\delta(\boldsymbol{y}) \right| \le L_f \|\boldsymbol{x} - \boldsymbol{y}\|.$$

*Proof.* From Definition 2.1 and (A2), we have, for all $\boldsymbol{x}, \boldsymbol{y} \in \mathbb{R}^d$,

$$
\begin{aligned}
\left| \hat{f}_\delta(\boldsymbol{x}) - \hat{f}_\delta(\boldsymbol{y}) \right| &= |\mathbb{E}_{\boldsymbol{u}}\left[f(\boldsymbol{x} - \delta\boldsymbol{u})\right] - \mathbb{E}_{\boldsymbol{u}}\left[f(\boldsymbol{y} - \delta\boldsymbol{u})\right]| \\
&= |\mathbb{E}_{\boldsymbol{u}}\left[f(\boldsymbol{x} - \delta\boldsymbol{u}) - f(\boldsymbol{y} - \delta\boldsymbol{u})\right]| \\
&\le \mathbb{E}_{\boldsymbol{u}}\left[|f(\boldsymbol{x} - \delta\boldsymbol{u}) - f(\boldsymbol{y} - \delta\boldsymbol{u})|\right] \\
&\le \mathbb{E}_{\boldsymbol{u}}\left[L_f \|(\boldsymbol{x} - \delta\boldsymbol{u}) - (\boldsymbol{y} - \delta\boldsymbol{u})\|\right] \\
&= \mathbb{E}_{\boldsymbol{u}}\left[L_f \|\boldsymbol{x} - \boldsymbol{y}\|\right] \\
&= L_f \|\boldsymbol{x} - \boldsymbol{y}\|.
\end{aligned}
$$

This completes the proof. $\qquad\square$

Lemmas D.1 and D.2 imply that the Lipschitz constants $L_f$ of the original function $f$ and $L_g$ of $\nabla f$ are taken over by the smoothed function $\hat{f}_\delta$ and its gradient $\nabla \hat{f}_\delta$ for all $\delta \in \mathbb{R}$.

# E   LEMMAS USED IN THE PROOFS OF THE THEOREMS

**Lemma E.1.** *Suppose that $\hat{f}_{\delta_m} \colon \mathbb{R}^d \to \mathbb{R}$ is $\sigma_m$-strongly convex and $\hat{\boldsymbol{x}}_{t+1}^{(m)} := \hat{\boldsymbol{x}}_t^{(m)} - \eta_t \boldsymbol{g}_t$. Then, for all $t \in \mathbb{N}$,*

$$\hat{f}_{\delta_m}(\hat{\boldsymbol{x}}_t^{(m)}) - \hat{f}_{\delta_m}(\boldsymbol{x}^\star) \le \frac{1 - \sigma_m \eta_t}{2\eta_t} X_t - \frac{1}{2\eta_t} X_{t+1} + \frac{\eta_t}{2}\|\boldsymbol{g}_t\|^2,$$

*where $\boldsymbol{g}_t := \nabla \hat{f}_{\delta_m}(\hat{\boldsymbol{x}}_t^{(m)})$, $X_t := \|\hat{\boldsymbol{x}}_t^{(m)} - \boldsymbol{x}_{\delta_m}^\star\|^2$, and $\boldsymbol{x}_{\delta_m}^\star$ is the global minimizer of $\hat{f}_{\delta_m}$.*

*Proof.* Let $t \in \mathbb{N}$. The definition of $\hat{\boldsymbol{x}}_{t+1}^{(m)}$ guarantees that

$$
\begin{aligned}
\|\hat{\boldsymbol{x}}_{t+1}^{(m)} - \boldsymbol{x}^\star\|^2 &= \|(\hat{\boldsymbol{x}}_t^{(m)} - \eta_t \boldsymbol{g}_t) - \boldsymbol{x}^\star\|^2 \\
&= \|\hat{\boldsymbol{x}}_t^{(m)} - \boldsymbol{x}^\star\|^2 - 2\eta_t \langle \hat{\boldsymbol{x}}_t^{(m)} - \boldsymbol{x}_{\delta_m}^\star, \boldsymbol{g}_t \rangle + \eta_t^2 \|\boldsymbol{g}_t\|^2.
\end{aligned}
$$

From the $\sigma_m$-strong convexity of $\hat{f}_{\delta_m}$,

$$\|\hat{\boldsymbol{x}}_{t+1}^{(m)} - \boldsymbol{x}_{\delta_m}^\star\|^2 \le \|\hat{\boldsymbol{x}}_t^{(m)} - \boldsymbol{x}_{\delta_m}^\star\|^2 + 2\eta_t \left( \hat{f}_{\delta_m}(\boldsymbol{x}_{\delta_m}^\star) - \hat{f}_{\delta_m}(\hat{\boldsymbol{x}}_t^{(m)}) - \frac{\sigma_m}{2}\|\hat{\boldsymbol{x}}_t^{(m)} - \boldsymbol{x}_{\delta_m}^\star\|^2 \right) + \eta_t^2 \|\boldsymbol{g}_t\|^2.$$

Hence,

$$\hat{f}_{\delta_m}(\hat{\boldsymbol{x}}_t^{(m)}) - \hat{f}_{\delta_m}(\boldsymbol{x}_{\delta_m}^\star) \le \frac{1 - \sigma_m \eta_t}{2\eta_t} \|\hat{\boldsymbol{x}}_t^{(m)} - \boldsymbol{x}_{\delta_m}^\star\|^2 - \frac{1}{2\eta_t} \|\hat{\boldsymbol{x}}_{t+1}^{(m)} - \boldsymbol{x}_{\delta_m}^\star\|^2 + \frac{\eta_t}{2} \|\boldsymbol{g}_t\|^2.$$

This completes the proof. $\qquad\square$

**Lemma E.2.** *Suppose that $\hat{f}_{\delta_m} : \mathbb{R}^d \to \mathbb{R}$ is $L_g$-smooth and $\hat{\boldsymbol{x}}_{t+1}^{(m)} := \hat{\boldsymbol{x}}_t^{(m)} - \eta_t \boldsymbol{g}_t$. Then, for all $t \in \mathbb{N}$,*

$$\eta_t \left(1 - \frac{L_g \eta_t}{2}\right) \|\nabla \hat{f}_{\delta_m}(\hat{\boldsymbol{x}}_t^{(m)})\|^2 \le \hat{f}_{\delta_m}(\hat{\boldsymbol{x}}_t^{(m)}) - \hat{f}_{\delta_m}(\hat{\boldsymbol{x}}_{t+1}^{(m)}).$$

*where $\boldsymbol{g}_t := \nabla \hat{f}_{\delta_m}(\hat{\boldsymbol{x}}_t^{(m)})$ and $\boldsymbol{x}_{\delta_m}^\star$ is the global minimizer of $\hat{f}_{\delta_m}$.*

*Proof.* From the $L_g$-smoothness of the $\hat{f}_{\delta_m}$ and the definition of $\hat{\boldsymbol{x}}_{t+1}^{(m)}$, we have, for all $t \in \mathbb{N}$,

$$\hat{f}_{\delta_m}(\hat{\boldsymbol{x}}_{t+1}^{(m)}) \le \hat{f}_{\delta_m}(\hat{\boldsymbol{x}}_t^{(m)}) + \langle \nabla \hat{f}_{\delta_m}(\hat{\boldsymbol{x}}_t^{(m)}), \hat{\boldsymbol{x}}_{t+1}^{(m)} - \hat{\boldsymbol{x}}_t^{(m)} \rangle + \frac{L_g}{2} \|\hat{\boldsymbol{x}}_{t+1}^{(m)} - \hat{\boldsymbol{x}}_t^{(m)}\|^2$$

$$= \hat{f}_{\delta_m}(\hat{\boldsymbol{x}}_t^{(m)}) - \eta_t \langle \nabla \hat{f}_{\delta_m}(\hat{\boldsymbol{x}}_t^{(m)}), \boldsymbol{g}_t \rangle + \frac{L_g \eta_t^2}{2} \|\boldsymbol{g}_t\|^2$$

$$\le \hat{f}_{\delta_m}(\hat{\boldsymbol{x}}_t^{(m)}) - \eta_t \left(1 - \frac{L_g \eta_t}{2}\right) \|\nabla \hat{f}_{\delta_m}(\hat{\boldsymbol{x}}_t^{(m)})\|^2.$$

Therefore, we have

$$\eta_t \left(1 - \frac{L_g \eta_t}{2}\right) \|\nabla \hat{f}_{\delta_m}(\hat{\boldsymbol{x}}_t^{(m)})\|^2 \le \hat{f}_{\delta_m}(\hat{\boldsymbol{x}}_t^{(m)}) - \hat{f}_{\delta_m}(\hat{\boldsymbol{x}}_{t+1}^{(m)}).$$

This completes the proof. $\qquad\square$

**Lemma E.3.** *Suppose that $\hat{f}_{\delta_m} : \mathbb{R}^d \to \mathbb{R}$ is $L_g$-smooth, $\hat{\boldsymbol{x}}_{t+1}^{(m)} := \hat{\boldsymbol{x}}_t^{(m)} - \eta_t \boldsymbol{g}_t$, and $\eta_t := \eta < \frac{2}{L_g}$. Then, for all $t \in \mathbb{N}$,*

$$\frac{1}{T} \sum_{t=1}^T \|\boldsymbol{g}_t\|^2 \le \frac{6 L_f \delta_m}{\eta(2 - L_g \eta) T},$$

*where $\boldsymbol{g}_t := \nabla \hat{f}_{\delta_m}(\hat{\boldsymbol{x}}_t^{(m)})$ and $\boldsymbol{x}_{\delta_m}^\star$ is the global minimizer of $\hat{f}_{\delta_m}$.*

*Proof.* According to Lemma E.2, we have

$$\eta \left(1 - \frac{L_g \eta}{2}\right) \|\nabla F(\boldsymbol{x}_t^{(m)})\|^2 \le \hat{f}_{\delta_m}(\hat{\boldsymbol{x}}_t^{(m)}) - \hat{f}_{\delta_m}(\hat{\boldsymbol{x}}_{t+1}^{(m)}).$$

Summing over $t$, we find that

$$\eta \left(1 - \frac{L_g \eta}{2}\right) \frac{1}{T} \sum_{t=1}^T \|\nabla \hat{f}_{\delta_m}(\hat{\boldsymbol{x}}_t^{(m)})\|^2 \le \frac{\hat{f}_{\delta_m}(\hat{\boldsymbol{x}}_1^{(m)}) - \hat{f}_{\delta_m}(\hat{\boldsymbol{x}}_{T+1}^{(m)})}{T}.$$

Hence, from $\eta < \frac{2}{L_g}$,

$$\frac{1}{T} \sum_{t=1}^T \|\boldsymbol{g}_t\|^2 = \frac{2 \left(\hat{f}_{\delta_m}(\hat{\boldsymbol{x}}_1^{(m)}) - \hat{f}_{\delta_m}(\boldsymbol{x}_{\delta_m}^\star)\right)}{\eta(2 - L_g \eta) T}.$$

Here, from the $L_f$-Lipschitz continuity of $\hat{f}_{\delta_m}$,

$$\hat{f}_{\delta_m}(\hat{\boldsymbol{x}}_1^{(m)}) - \hat{f}_{\delta_m}(\boldsymbol{x}_{\delta_m}^\star) \le L_f \|\hat{\boldsymbol{x}}_1^{(m)} - \boldsymbol{x}_{\delta_m}^\star\|$$
$$\le 3 L_f \delta_m,$$

where we have used $\boldsymbol{x}_1^{(m)} \in B(\boldsymbol{x}_{\delta_m}^\star; 3\delta_m)$. Therefore, we have

$$\frac{1}{T} \sum_{t=1}^T \|\boldsymbol{g}_t\|^2 = \frac{6 L_f \delta_m}{\eta(2 - L_g \eta) T}.$$

This completes the proof. $\qquad\square$

## F   PROOF OF THE THEOREMS AND PROPOSITIONS

### F.1   PROOF OF THEOREM 5.1

*Proof.* Lemma E.1 guarantees that

$$\hat{f}_{\delta_m}(\hat{\boldsymbol{x}}_t^{(m)}) - \hat{f}_{\delta_m}(\boldsymbol{x}_{\delta_m}^\star) \leq \frac{1 - \sigma_m \eta_t}{2\eta_t} X_t - \frac{1}{2\eta_t} X_{t+1} + \frac{\eta_t}{2}\|\boldsymbol{g}_t\|^2$$

$$= \frac{1 - \sigma_m \eta}{2\eta}(X_t - X_{t+1}) - \frac{\sigma_m}{2} X_{t+1} + \frac{\eta}{2}\|\boldsymbol{g}_t\|^2.$$

From $\eta < \min\left\{\frac{1}{\sigma_m}, \frac{2}{L_g}\right\}$ and Lemma E.3, by summing over $t$ we find that

$$\frac{1}{T}\sum_{t=1}^{T}\left(\hat{f}_{\delta_m}(\hat{\boldsymbol{x}}_t^{(m)}) - \hat{f}_{\delta_m}(\boldsymbol{x}_{\delta_m}^\star)\right) \leq \frac{1 - \sigma_m \eta}{2\eta T}(X_1 - X_{T+1}) - \frac{\sigma_m}{2T}\sum_{t=1}^{T}X_{t+1} + \frac{\eta}{2T}\sum_{t=1}^{T}\|\boldsymbol{g}_t\|^2$$

$$\leq \frac{1 - \sigma_m \eta}{2\eta T}X_1 + \frac{\eta}{2T}\sum_{t=1}^{T}\|\boldsymbol{g}_t\|^2$$

$$\leq \underbrace{\frac{9(1 - \sigma_m \eta)\delta_m^2}{2\eta}}_{=:H_1}\frac{1}{T} + \underbrace{\frac{3L_f \delta_m}{\eta(2 - L_g \eta)}}_{=:H_2}\frac{1}{T}$$

$$= \underbrace{(H_1 + H_2)}_{=:H_m}\frac{1}{T}$$

$$= \frac{H_m}{T},$$

where we have used $X_1 := \|\hat{\boldsymbol{x}}_1^{(m)} - \boldsymbol{x}_{\delta_m}^\star\|^2 \leq 9\delta_m^2$ and $H_m > 0$ is a nonnegative constant. From the convexity of $F$,

$$\hat{f}_{\delta_m}\left(\frac{1}{T}\sum_{t=1}^{T}\hat{\boldsymbol{x}}_t^{(m)}\right) \leq \frac{1}{T}\sum_{t=1}^{T}\hat{f}_{\delta_m}(\hat{\boldsymbol{x}}_t^{(m)}).$$

Hence,

$$\hat{f}_{\delta_m}\left(\frac{1}{T}\sum_{t=1}^{T}\hat{\boldsymbol{x}}_t^{(m)}\right) - \hat{f}_{\delta_m}(\boldsymbol{x}_{\delta_m}^\star) \leq \frac{H_m}{T} = \mathcal{O}\left(\frac{1}{T}\right).$$

In addition, since the minimum value is smaller than the mean, we have

$$\min_{t \in [T]}\left(\hat{f}_{\delta_m}\left(\hat{\boldsymbol{x}}_t^{(m)}\right) - \hat{f}_{\delta_m}(\boldsymbol{x}_{\delta_m}^\star)\right) \leq \frac{H_m}{T} = \mathcal{O}\left(\frac{1}{T}\right).$$

This completes the proof. □

### F.2   PROOF OF PROPOSITION 5.1

*Proof.* This proposition can be proved by induction. Since we assume $\boldsymbol{x}_1 \in N(\boldsymbol{x}_{\delta_1}^\star; 3\delta_1)$, we have

$$\|\boldsymbol{x}_1 - \boldsymbol{x}_{\delta_1}^\star\| < 3\delta_1,$$

which establishes the case of $m = 1$. Now let us assume that the proposition holds for any $m > 1$. Accordingly, the initial point $\boldsymbol{x}_m$ for the optimization of the $m$-th smoothed function $\hat{f}_{\delta_m}$ and its global optimal solution $\boldsymbol{x}_{\delta_m}^\star$ are both contained in the its $\sigma_m$-strongly convex region $N(\boldsymbol{x}_{\delta_m}^\star; 3\delta_m)$. Thus, after $T_m := H_m/\epsilon_m$ iterations, Algorithm 2 (GD) returns an approximate solution $\hat{\boldsymbol{x}}_{T_m+1}^{(m)} =: \boldsymbol{x}_{m+1}$, and the following holds from Theorem 5.1:

$$\hat{f}_{\delta_m}(\boldsymbol{x}_{m+1}) - \hat{f}_{\delta_m}(\boldsymbol{x}_{\delta_m}^\star) \leq \frac{H_m}{T_m} = \epsilon_m := \frac{\sigma_m \delta_m^2}{2} = \frac{\sigma_m \delta_{m+1}^2}{2\gamma^2}.$$

Hence, from the $\sigma_m$-strongly convexity of $\hat{f}_{\delta_m}$,

$$\frac{\sigma_m}{2}\|\boldsymbol{x}_{m+1} - \boldsymbol{x}^\star_{\delta_m}\|^2 \leq \frac{\sigma_m \delta^2_{m+1}}{2\gamma^2}, \text{ i.e., } \|\boldsymbol{x}_{m+1} - \boldsymbol{x}^\star_{\delta_m}\| \leq \frac{\delta_{m+1}}{\gamma}$$

Therefore, from the $\sigma_m$-niceness of $f$ and $\gamma \in [0.5, 1)$,

$$\|\boldsymbol{x}_{m+1} - \boldsymbol{x}^\star_{\delta_{m+1}}\| \leq \|\boldsymbol{x}_{m+1} - \boldsymbol{x}^\star_{\delta_m}\| + \|\boldsymbol{x}^\star_{\delta_m} - \boldsymbol{x}^\star_{\delta_{m+1}}\|$$
$$\leq \frac{\delta_{m+1}}{\gamma} + (|\delta_m| - \delta_{m+1})$$
$$= \frac{\delta_{m+1}}{\gamma} + \left(\frac{\delta_{m+1}}{\gamma} - \delta_{m+1}\right)$$
$$= \left(\frac{2}{\gamma} - 1\right)\delta_{m+1}$$
$$\leq 3\delta_{m+1}.$$

This completes the proof. $\qquad\square$

### F.3    PROOF OF THEOREM 5.2

The following proof uses the technique presented in (Hazan et al., 2016).

*Proof.* According to $\delta_{m+1} := \frac{\eta_{m+1}C}{\sqrt{b_{m+1}}}$ and $\frac{\kappa_m}{\sqrt{\lambda_m}} = \gamma$, we have

$$\delta_{m+1} := \frac{\eta_{m+1}C}{\sqrt{b_{m+1}}}$$
$$= \frac{\kappa_m \eta_m C}{\sqrt{\lambda_m}\sqrt{b_m}}$$
$$= \frac{\kappa_m}{\sqrt{\lambda_m}}\delta_m$$
$$= \gamma\delta_m.$$

Therefore, from $M := \log_\gamma(\alpha_0\epsilon) + 1$ and $\delta_1 := \frac{\eta_1 C}{\sqrt{b_1}}$

$$\delta_M = \delta_1 \gamma^{M-1}$$
$$= \delta_1 \alpha_0 \epsilon$$
$$= \frac{\eta_1 C \alpha_0 \epsilon}{\sqrt{b_1}}.$$

According to Theorem 5.1,

$$\mathbb{E}\left[\hat{f}_{\delta_M}(\boldsymbol{x}_{M+1}) - \hat{f}_{\delta_M}(\boldsymbol{x}^\star_{\delta_M})\right] \leq \epsilon_M$$
$$= \sigma_M \delta^2_M$$
$$= \left(\frac{\sqrt{\sigma_M}\eta_1 C \alpha_0 \epsilon}{\sqrt{b_1}}\right)^2$$

From Lemmas D.2 and 2.2,

$$f(\boldsymbol{x}_{M+2}) - f(\boldsymbol{x}^\star) = \left\{f(\boldsymbol{x}_{M+2}) - \hat{f}_{\delta_M}(\boldsymbol{x}_{M+2})\right\} + \left\{\hat{f}_{\delta_M}(\boldsymbol{x}^\star) - f(\boldsymbol{x}^\star)\right\} + \left\{\hat{f}_{\delta_M}(\boldsymbol{x}_{M+2}) - \hat{f}_{\delta_M}(\boldsymbol{x}^\star)\right\}$$
$$\leq \left\{f(\boldsymbol{x}_{M+2}) - \hat{f}_{\delta_M}(\boldsymbol{x}_{M+2})\right\} + \left\{\hat{f}_{\delta_M}(\boldsymbol{x}^\star) - f(\boldsymbol{x}^\star)\right\} + \left\{\hat{f}_{\delta_M}(\boldsymbol{x}_{M+2}) - \hat{f}_{\delta_M}(\boldsymbol{x}^\star_{\delta_M})\right\}$$
$$\leq \delta_M L_f + \delta_M L_f + \left\{\hat{f}_{\delta_M}(\boldsymbol{x}_{M+2}) - \hat{f}_{\delta_M}(\boldsymbol{x}^\star_{\delta_M})\right\}$$
$$= 2\delta_M L_f + \left\{\hat{f}_{\delta_M}(\boldsymbol{x}_{M+2}) - \hat{f}_{\delta_M}(\boldsymbol{x}_{M+1})\right\} + \left\{\hat{f}_{\delta_M}(\boldsymbol{x}_{M+1}) - \hat{f}_{\delta_M}(\boldsymbol{x}^\star_{\delta_M})\right\}$$
$$\leq 2\delta_M L_f + L_f \|\boldsymbol{x}_{M+2} - \boldsymbol{x}_{M+1}\| + \left\{\hat{f}_{\delta_M}(\boldsymbol{x}_{M+1}) - \hat{f}_{\delta_M}(\boldsymbol{x}^\star_{\delta_M})\right\}.$$

Then, we have

$$f(\boldsymbol{x}_{M+2}) - f(\boldsymbol{x}^\star) \leq 2\delta_M L_f + 6L_f \delta_M + \epsilon_M$$
$$= 8L_f \delta_M + \epsilon_M,$$

where we have used $\|\boldsymbol{x}_{M+2} - \boldsymbol{x}_{M+1}\| \leq 6\delta_M$ since $\boldsymbol{x}_{M+2}, \boldsymbol{x}_{M+1} \in N(\boldsymbol{x}^\star; 3\delta_M)$. Therefore,

$$f(\boldsymbol{x}_{M+2}) - f(\boldsymbol{x}^\star) \leq \frac{8L_f \eta_1 C \alpha_0 \epsilon}{\sqrt{b_1}} + \left(\frac{\sqrt{\sigma_M} \eta_1 C \alpha_0 \epsilon}{\sqrt{b_1}}\right)^2$$

$$\leq \frac{8L_f \eta_1 C \alpha_0 \epsilon}{\sqrt{b_1}} + \left(\frac{\sqrt{\sigma} \eta_1 C \alpha_0 \epsilon}{\sqrt{b_1}}\right)^2$$

$$\leq \epsilon,$$

where we have used $\alpha_0 := \min\left\{\frac{\sqrt{b_1}}{16L_f \eta_1 C}, \frac{\sqrt{b_1}}{\sqrt{2\sigma}\eta_1 C}\right\}$.

Let $T_{\text{total}}$ be the total number of queries made by Algorithm 1; then,

$$T_{\text{total}} = \sum_{m=1}^{M+1} \frac{H_m}{\epsilon_m} = \sum_{m=1}^{M+1} \frac{H_m}{\sigma \delta_m^2}.$$

Here, from the proof of Theorem 5.1 (see Section F.1), we define $H_4 > 0$ as follows:

$$H_m := \frac{9(1 - \sigma_m \eta)\delta_m^2}{2\eta} + \frac{3L_f \delta_m}{\eta(2 - L_g \eta)} \leq \frac{9(1 - \sigma_1 \eta)\delta_1^2}{2\eta} + \frac{3L_f \delta_1}{\eta(2 - L_g \eta)} =: H_4$$

Thus, from $\delta_M = \delta_1 \alpha_0 \epsilon$,

$$T_{\text{total}} = \sum_{m=1}^{M+1} \frac{H_m}{\sigma_m \delta_m^2} \leq H_4 \sum_{m=1}^{M+1} \frac{1}{\sigma_m \delta_m^2} \leq H_4 \sum_{m=1}^{M+1} \frac{1}{\sigma_1 \delta_M^2} = \frac{H_4(M+1)}{\sigma_1 \delta_M^2}$$

$$= \frac{H_4(M+1)}{\sigma_1 \delta_1^2 \alpha_0^2 \epsilon^2} = \mathcal{O}\left(\frac{1}{\epsilon^2}\right).$$

This completes the proof. $\square$

## G  FULL EXPERIMENTAL RESULTS

The experimental environment was as follows: NVIDIA GeForce RTX 4090×2GPU and Intel Core i9 13900KF CPU. The software environment was Python 3.10.12, PyTorch 2.1.0 and CUDA 12.2.

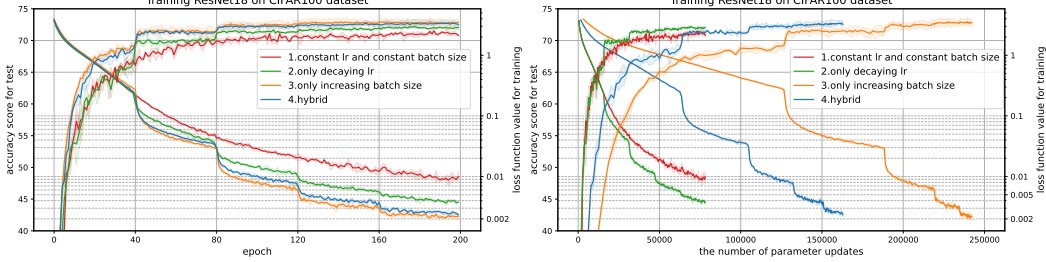

Figure 5: Accuracy score for testing and loss function value for training versus the number of epochs (**left**) and the number of parameter updates (**right**) in training ResNet18 on the CIFAR100 dataset. The solid line represents the mean value, and the shaded area represents the maximum and minimum over three runs. In method 1, the learning rate and the batch size were fixed at 0.1 and 128, respectively. In method 2, the learning rate decreased every 40 epochs as $\left[0.1, \frac{1}{10\sqrt{2}}, 0.05, \frac{1}{20\sqrt{2}}, 0.025\right]$ and the batch size was fixed at 128. In method 3, the learning rate was fixed at 0.1, and the batch size was increased as $[16, 32, 64, 128, 256]$. In method 4, the learning rate was decreased as $\left[0.1, \frac{\sqrt{3}}{20}, 0.075, \frac{3\sqrt{3}}{80}, 0.05625\right]$ and the batch size was increased as $[32, 48, 72, 108, 162]$.

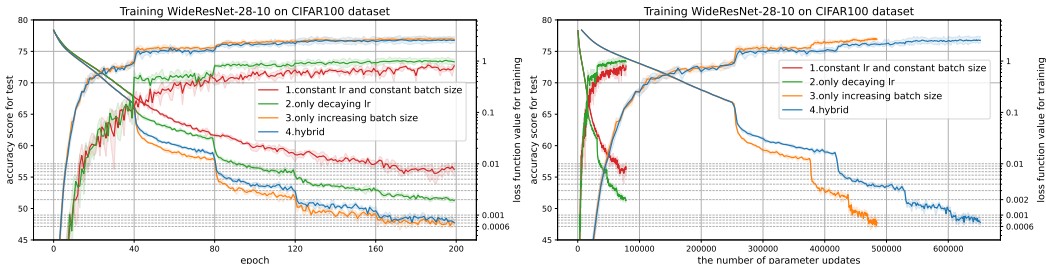

Figure 6: Accuracy score for testing and loss function value for training versus the number of epochs (**left**) and the number of parameter updates (**right**) in training WideResNet-28-10 on the CIFAR100 dataset. The solid line represents the mean value, and the shaded area represents the maximum and minimum over three runs. In method 1, the learning rate and batch size were fixed at 0.1 and 128, respectively. In method 2, the learning rate was decreased every 40 epochs as $\left[0.1, \frac{1}{10\sqrt{2}}, 0.05, \frac{1}{20\sqrt{2}}, 0.025\right]$ and the batch size was fixed at 128. In method 3, the learning rate was fixed at 0.1, and the batch size was increased as $[8, 16, 32, 64, 128]$. In method 4, the learning rate was decreased as $\left[0.1, \frac{\sqrt{3}}{20}, 0.075, \frac{3\sqrt{3}}{80}, 0.05625\right]$ and the batch size was increased as $[8, 12, 18, 27, 40]$.

For the sake of fairness, we provide here a version of Figures 4-6 with the number of gradient queries on the horizontal axis (see Figure 7-9). Since $b$ stochastic gradients are computed per epoch, the number of gradient queries is $Tb$, where $T$ means the number of steps and $b$ means the batch size.

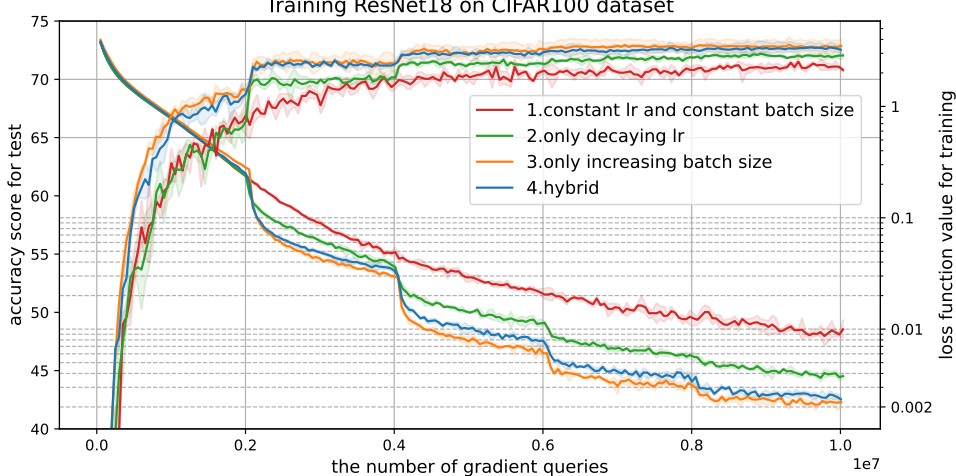

Figure 7: Accuracy score for testing and loss function value for training versus the number of gradient queries in training ResNet18 on the CIFAR100 dataset. The solid line represents the mean value, and the shaded area represents the maximum and minimum over three runs. In method 1, the learning rate and the batch size were fixed at 0.1 and 128, respectively. In method 2, the learning rate decreased every 40 epochs as in $\left[0.1, \frac{1}{10\sqrt{2}}, 0.05, \frac{1}{20\sqrt{2}}, 0.025\right]$ and the batch size was fixed at 128. In method 3, the learning rate was fixed at 0.1, and the batch size was increased as $[16, 32, 64, 128, 256]$. In method 4, the learning rate was decreased as $\left[0.1, \frac{\sqrt{3}}{20}, 0.075, \frac{3\sqrt{3}}{80}, 0.05625\right]$ and the batch size was increased as $[32, 48, 72, 108, 162]$. **This graph shows almost the same results as Figure 5.**

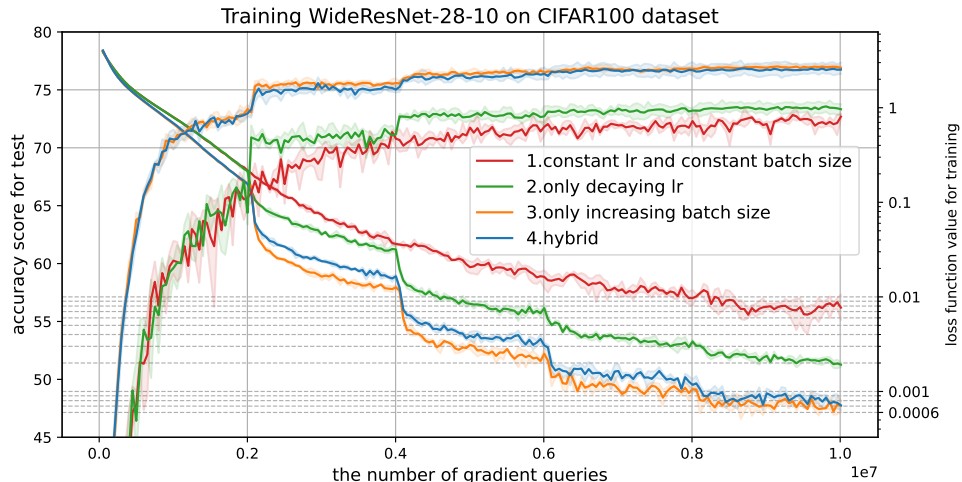

Figure 8: Accuracy score for testing and loss function value for training versus the number of gradient queries in training WideResNet-28-10 on the CIFAR100 dataset. The solid line represents the mean value, and the shaded area represents the maximum and minimum over three runs. In method 1, the learning rate and batch size were fixed at 0.1 and 128, respectively. In method 2, the learning rate was decreased every 40 epochs as $\left[0.1, \frac{1}{10\sqrt{2}}, 0.05, \frac{1}{20\sqrt{2}}, 0.025\right]$ and the batch size was fixed at 128. In method 3, the learning rate was fixed at 0.1, and the batch size increased as $[8, 16, 32, 64, 128]$. In method 4, the learning rate decreased as $\left[0.1, \frac{\sqrt{3}}{20}, 0.075, \frac{3\sqrt{3}}{80}, 0.05625\right]$ and the batch size increased as $[8, 12, 18, 27, 40]$. **This graph shows almost the same results as Figure 6.**

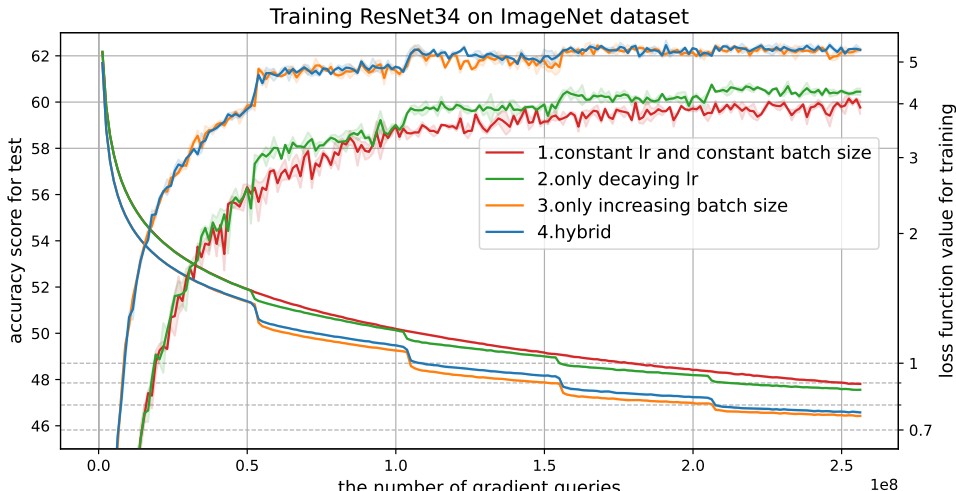

Figure 9: Accuracy score for testing and loss function value for training versus the number of gradient queries in training ResNet34 on the ImageNet dataset. The solid line represents the mean value, and the shaded area represents the maximum and minimum over three runs. In method 1, the learning rate and batch size were fixed at 0.1 and 256, respectively. In method 2, the learning rate was decreased every 40 epochs as $\left[0.1, \frac{1}{10\sqrt{2}}, 0.05, \frac{1}{20\sqrt{2}}, 0.025\right]$ and the batch size was fixed at 256. In method 3, the learning rate was fixed at 0.1, and the batch size was increased as $[32, 64, 128, 256, 512]$. In method 4, the learning rate was decreased as $\left[0.1, \frac{\sqrt{3}}{20}, 0.075, \frac{3\sqrt{3}}{80}, 0.05625\right]$ and the batch size was increased as $[32, 48, 72, 108, 162]$. **This graph shows almost the same results as Figure 4.**

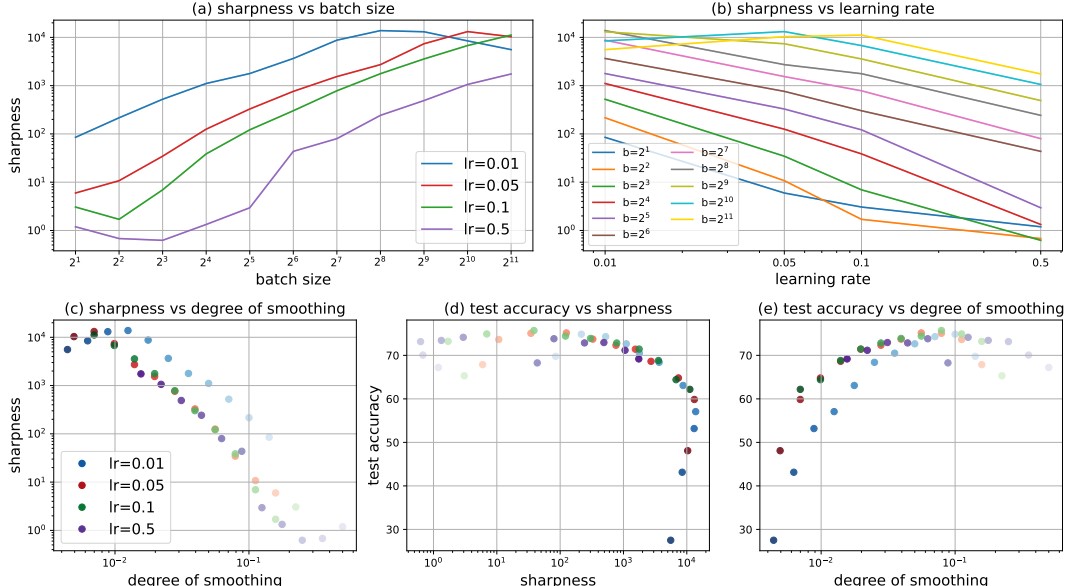

Figure 10: **(a)** Sharpness around the approximate solution after 200 epochs of WideResNet-28-10 training on the CIFAR100 dataset versus batch size used. **(b)** Sharpness versus learning rate used. **(c)** Sharpness versus degree of smoothing calculated from learning rate, batch size and estimated variance of the stochastic gradient. **(d)** Test accuracy after 200 epochs training versus sharpness. **(e)** Test accuracy versus degree of smoothing. The color shade in the scatter plots represents the batch size; the larger the batch size, the darker the color of the plotted points. "lr" means learning rate. The experimental results that make up the all graphs are all identical. **Dear Reviewers: These are plots of the results of a single experiment. We can provide the results of multiple experiments before camera ready.**

## H DISCUSSION ON THE DEFINITION OF THE SMOOTHED FUNCTION

Recall the general definition of the smoothing of the function.

**Definition H.1.** *Given a function $f\colon \mathbb{R}^d \to \mathbb{R}$, define $\hat{f}_\delta\colon \mathbb{R}^d \to \mathbb{R}$ to be the function obtained by smoothing $f$ as*

$$\hat{f}_\delta(\boldsymbol{x}) := \mathbb{E}_{\boldsymbol{u}\sim\mathcal{N}\left(\boldsymbol{0};\frac{1}{\sqrt{d}}I_d\right)}\left[f(\boldsymbol{x} - \delta\boldsymbol{u})\right],$$

*where $\delta > 0$ represents the degree of smoothing and $\boldsymbol{u}$ is a random variable from a Gaussian distribution.*

What probability distribution the random variable $\boldsymbol{u} \in \mathbb{R}^d$ follows in the definition of smoothed function varies in the literature. (Wu, 1996; Mobahi & Fisher III, 2015b; Iwakiri et al., 2022) assumes a Gaussian distribution, while (Hazan et al., 2016) assumes a uniform distribution for the sake of theoretical analysis. So what probability distribution the random variable $\boldsymbol{u}$ should follow in order to smooth the function? This has never been discussed.

It is difficult to confirm from a strictly theoretical point of view whether the function $\hat{f}_\delta$ obtained by using a random variable $\boldsymbol{u}$ that follows a certain probability distribution is smoother than the original function $f$ (more precisely, it is possible with a Gaussian distribution). Therefore, we smoothed a very simple nonconvex function with random variables following several major probability distributions and compared it with the original function. We deal with one-dimensional Rastrigin's function (Törn & Zilinskas, 1989; Rudolph, 1990) and Drop-Wave function (Marcin Molga, 2005) defined as follows:

$$\text{(Rastrigin's function)} \quad f(x) := x^2 - 10\cos(2\pi x) + 10, \tag{5}$$

$$\text{(Drop-Wave function)} \quad f(x) := -\frac{1 + \cos(12\pi x)}{0.5x^2 + 2}. \tag{6}$$

We smooth the above functions according to Definition 2.1 using random variables following light-tailed distributions: Gaussian, uniform, exponential, and Rayleigh, and heavy-tailed distributions: Pareto, Cauchy, and Levy. We have added the code for this smoothing experiment to our anonymous Github. For more information on the parameters of each probability distribution, please see there. First, Figure 11 plots the Rastrigin's function and its smoothed version with a degree of smoothing of $\delta = 0.5$, using a random variable that follows several probability distributions.

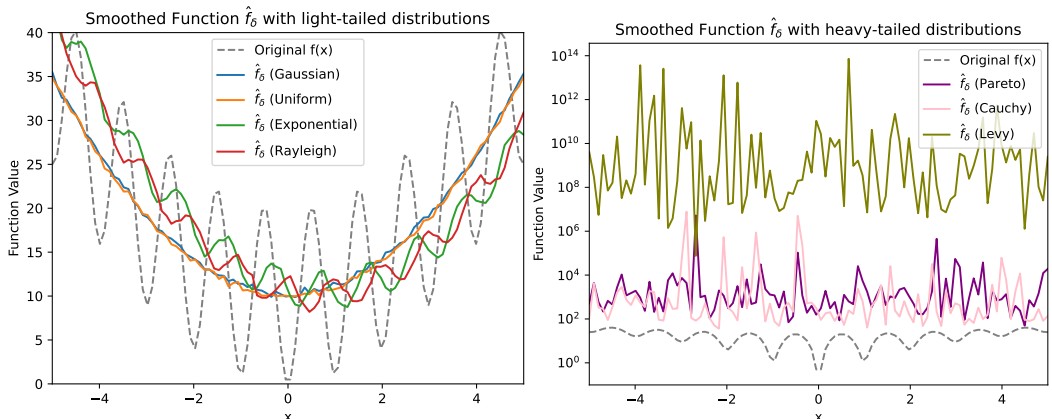

Figure 11: Rastrigin's function (5) and its smoothed version using random variables following a light-tailed distribution (**left**) and heavy-tailed distribution (**right**). The degree of smoothing is set to 0.5. Note that right graph has the logarithmic vertical axis with a base of 10.

Figures 11 shows that smoothing using random variable from light-tailed distributions works, while smoothing using random variable from heavy-tailed distributions does not. The reason for this is thought to be that extremely large values tend to appear in heavy-tailed distributions, and the function values are not stable.

Next, Figure 12 plots the Drop-Wave function and its smoothed version with a degree of smoothing of $\delta = 0.5$, using a random variable that follows several probability distributions.

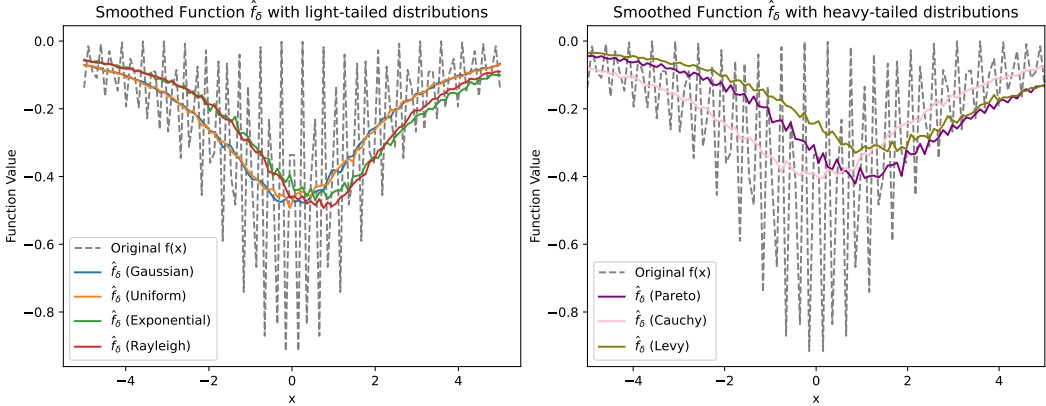

Figure 12: Drop-Wave function (6) and its smoothed version using random variables following a light-tailed distribution (**left**) and heavy-tailed distribution (**right**). The degree of smoothing is set to 0.5.

Figure 12 shows that, in contrast to Figure 11, the heavy-tailed distribution successfully smooths the function as well as the light-tailed distribution. The reason for this lies in the definition of the Drop-Wave function. The Drop-Wave function has an $x^2$ term in its denominator, which prevents the function value from exploding even when the heavy-tailed distribution provides extremely large values, and thus the smoothing works.

In smoothing of the function, random variables have been defined to follow primarily a Gaussian distribution (see Definition H.1), but these experimental results motivate us to extend it from a

Gaussian distribution to a light-tailed distribution (see Definition 2.1). Note that we also provide the interesting finding that, depending on the definition of the original function, random variables from heavy-tailed distributions can also be useful for smoothing.

## H.1 DISCUSSION ON THE LIGHT-TAILED DISTRIBUTION

According to (L.A. et al., 2020, Definition 3.1), the light-tailed distribution is defined as follows:

**Definition H.2** (light-tailed distribution). *A random variable $X$ is said to be light-tailed if there exists a $c_0 > 0$ such that $\mathbb{E}\left[\exp(\lambda X)\right] < \infty$ for all $|\lambda| < c_0$.*

This definition implies that the probability density function of the light-tailed distribution decreases exponentially at the tail. The definition of function smoothing in the previous study (Hazan et al., 2016, Definition 4.1) used a random variable that follows a uniform distribution. We can show that the uniform distribution is light-tailed distribution.

**Proposition H.1.** *The uniform distribution is light-tailed distribution.*

*Proof.* Let X be a random variable which follows uniform distribution over the interval $[a, b]$, where $a, b \in \mathbb{R}$ $(a \le b)$. Then, the probability density function $f_X(x)$ is defined as follows:

$$f_X(x) = \begin{cases} \frac{1}{b-a}, & \text{if } x \in [a, b], \\ 0, & \text{otherwise.} \end{cases}$$

For all $\lambda \in \mathbb{R} \setminus \{0\}$, we have

$$\mathbb{E}\left[e^{\lambda X}\right] = \int_a^b e^{\lambda x} f_X(x) dx = \frac{1}{b-a} \int_a^b e^{\lambda x} dx = \frac{1}{b-a} \cdot \frac{1}{\lambda} \left[e^{\lambda x}\right]_a^b = \frac{1}{\lambda(b-a)} \left(e^{\lambda b} - e^{\lambda a}\right).$$

When $\lambda = 0$, we have

$$\mathbb{E}\left[e^{\lambda X}\right] = \mathbb{E}\left[e^0\right] = 1.$$

Therefore, for all $\lambda \in \mathbb{R}$, we obtain $\mathbb{E}\left[e^{\lambda X}\right] < \infty$. $\qquad\square$

## H.2 DISTRIBUTION OF SGD'S STOCHASTIC NOISE

We collected 1000 each of stochastic noise $\boldsymbol{\omega}_t := \nabla f_{\mathcal{S}_t}(\boldsymbol{x}_t) - \nabla f(\boldsymbol{x}_t)$ and tested whether each element follows a light-tailed distribution. They were collected at the point where ResNet18 had been trained on the CIFAR100 dataset (10,000 steps). The code used in this experiment is available on our anonymous Github. ResNet18 has about 11M parameters, so $\boldsymbol{\omega}_t$ form an 11M-dimensional vector. Figure 13 plots the results for the $\boldsymbol{\omega}_t$ elements from dimension 0 to dimension 100,000. Figure 14 present the results for all elements. These results demonstrate that the SGD's stochastic noise $\boldsymbol{\omega}_t$ follows a light-tailed distribution.

## I  EXTENSION FROM $\sigma$-NICE FUNCTION TO $\sigma_m$-NICE FUNCTION

Hazan et al., proposed $\sigma$-nice function to analyze graduated optimization algorithm.

**Definition I.1** ($\sigma$-nice function (Hazan et al., 2016)). *Let $M \in \mathbb{N}$ and $m \in [M]$. A function $f: \mathbb{R}^d \to \mathbb{R}$ is said to be $\sigma$-nice if the following two conditions hold:*

*(i) For all $\delta_m > 0$ and all $\boldsymbol{x}_{\delta_m}^\star$, there exists $\boldsymbol{x}_{\delta_{m+1}}^\star$ such that: $\left\| \boldsymbol{x}_{\delta_m}^\star - \boldsymbol{x}_{\delta_{m+1}}^\star \right\| \le \delta_{m+1} := \frac{\delta_m}{2}$.*

*(ii) For all $\delta_m > 0$, the function $\hat{f}_{\delta_m}(\boldsymbol{x})$ over $N(\boldsymbol{x}_{\delta_m}^\star; 3\delta_m)$ is $\sigma$-strongly convex.*

Recall our $\sigma_m$-nice function (Definition 5.1).

**Definition I.2** ($\sigma_m$-nice function). *Let $M \in \mathbb{N}$, $m \in [M]$, and $\gamma \in [0.5, 1)$. A function $f: \mathbb{R}^d \to \mathbb{R}$ is said to be $\sigma_m$-nice if the following two conditions hold:*

*(i) For all $\delta_m > 0$ and all $\boldsymbol{x}_{\delta_m}^\star$, there exists $\boldsymbol{x}_{\delta_{m+1}}^\star$ such that: $\left\| \boldsymbol{x}_{\delta_m}^\star - \boldsymbol{x}_{\delta_{m+1}}^\star \right\| \le \delta_{m+1} := \gamma \delta_m$.*

*(ii) For all $\delta_m > 0$, the function $\hat{f}_{\delta_m}(\boldsymbol{x})$ over $N(\boldsymbol{x}_{\delta_m}^\star; 3\delta_m)$ is $\sigma_m$-strongly convex.*

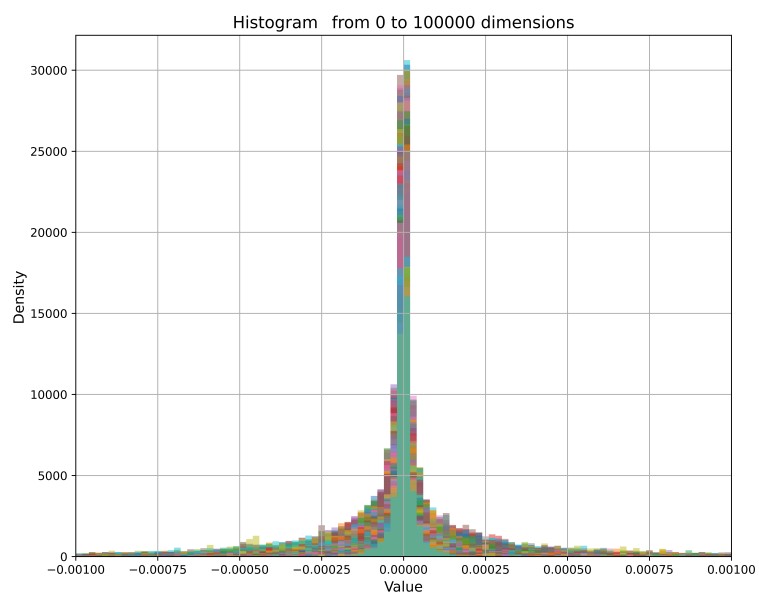

Figure 13: Distribution of 1000 $\boldsymbol{\omega}_t$ elements from 0 to 100,000 dimensions.

In condition (i), we extended the decay rate of the degree of smoothing from a constant 0.5 to a constant $\gamma \in [0.5, 1)$. See Proposition 5.1 for the soundness of this extension. In condition (ii), $\hat{f}_{\delta_m}$ was always defined to be $\sigma$-strongly convex in the definition of $\sigma$-nice function, which is a rather strong assumption. In fact, the greater the degree of smoothing $\delta_m$ is, the smoother the smoothed function $\hat{f}_{\delta_m}$ becomes and the smaller the strong convexity parameter may be. Here, let $\sigma_{\text{small}}$ be a strongly convexity parameter of $f$ and $0 < \sigma_{\text{small}} < \sigma_{\text{big}}$, then the function $f$ is not $\sigma_{\text{big}}$-strongly convex function. Therefore, the strongly convexity parameter should depend on the degree of smoothing $\delta_m$ and we extend the strongly convexity parameter to $\sigma_m$ from $\sigma$.

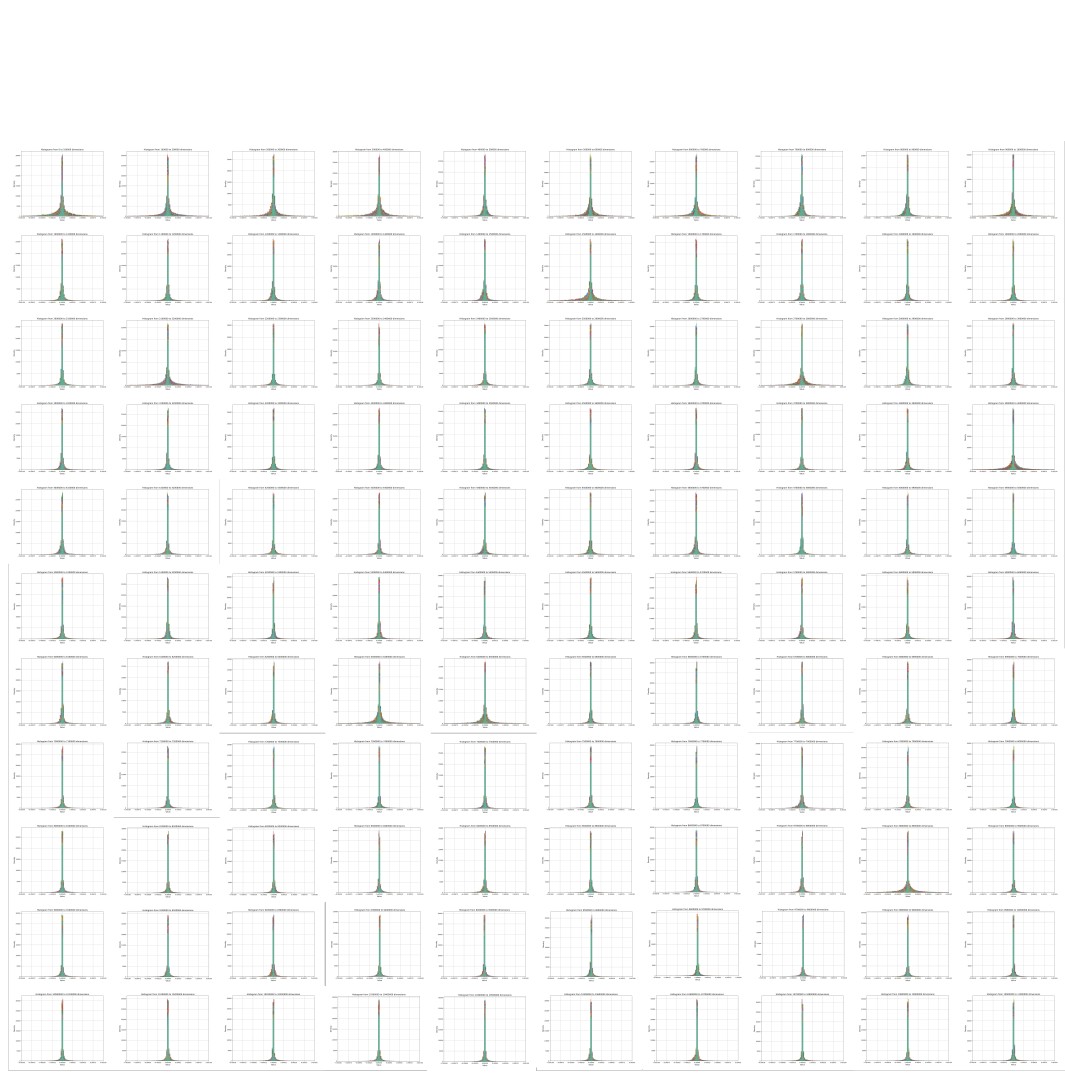

Figure 14: Complete results for distribution of 1000 $\omega_t$ elements. The distribution is plotted separately for each 100,000 dimensions.

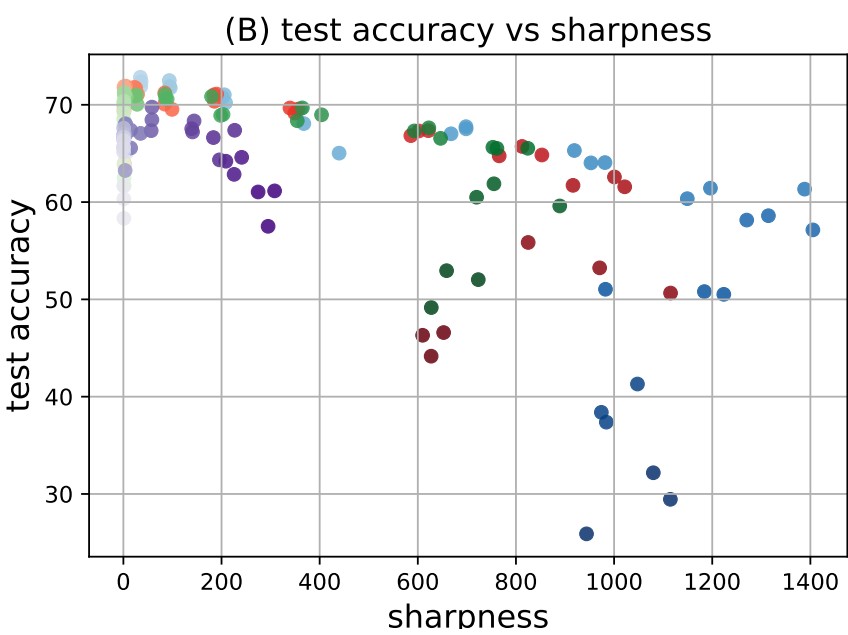

Figure 15: Graph for reviewer KvGj.

