# OpenReview forum: "Using Stochastic Gradient Descent to Smooth Nonconvex Functions: Analysis of Implicit Graduated Optimization"
_ICLR.cc/2025/Conference — Submitted to ICLR 2025_

### Official Review · Reviewer_kxU4 · 2024-11-01

**Soundness:** 2
**Presentation:** 2
**Contribution:** 2
**Rating:** 5
**Confidence:** 3

**Summary:**

The authors first analyzed the relationship between the batch size, learning rate, and test accuracy, showing that there is a correlation between $\frac{\eta C}{\sqrt{b}}$ and test accuracy.
Then, using these observations, the authors proposed Implicit Graduated Optimization, which changes the learning rate and batch size during the training.
The authors provided the convergence rate of Implicit Graduated Optimization and experimentally examined the effectiveness of their proposed method.

**Strengths:**

* The authors analyzed the relationship between test accuracy, learning rate, and batch size.

* Based on this relationship, the authors proposed Implicit Graduated Optimization that adjusts the batch sizes and learning rate during the training.

**Weaknesses:**

Overall, the reviewer feels that the proposed method itself is similar to that presented in previous studies, e.g., [1], and the clear advantage of the proposed methods over [1] has not been shown in this paper.
Designing the scheduler of batch sizes and learning rates from the perspective of graduated optimization seems to be novel, while the reviewer feels that the relationship between test accuracy and $\frac{\eta C}{\sqrt{b}}$, derived as a conclusion, does not appear to be very novel.
See below for a detailed comment.


* The reviewer does not think the relationship between $\frac{\eta C}{\sqrt{b}}$ and accuracy presented in this paper is very new since it is well-known that large batch sizes and small learning rates degrade test accuracy. While showing this relationship is a good motivation for designing the proposed method, the reviewer does not think that showing this relationship in itself is a major contribution.

* The reviewer does not understand the difference between the proposed method and existing methods, e.g., [1]. Changing the batch size during training has already been proposed in [1].

* All methods achieved approximately 60% in Figure 4. However, by comparing the results reported in the existing papers [1,2], 60% appears to be too low. Thus, the reviewer is wondering if the results are reliable.

### Typo
* "." is missing in "Similar early approaches can be found in (Witkin et al., 1987) and (Yuille, 1989)" in line 67.

## Reference

[1] Samuel et al., Don't Decay the Learning Rate, Increase the Batch Size, In ICLR 2018

[2] He et al., Deep Residual Learning for Image Recognition, In CVPR 2016

**Questions:**

See the weakness section.

---

> ### Author Response · Authors · 2024-11-14
>
> **Weaknesses 1 and 2:** The reviewer does not think the relationship between $\eta C/\sqrt{b}$ and accuracy presented in this paper is very new since it is well-known that large batch sizes and small learning rates degrade test accuracy. While showing this relationship is a good motivation for designing the proposed method, the reviewer does not think that showing this relationship in itself is a major contribution. The reviewer does not understand the difference between the proposed method and existing methods, e.g., [1]. Changing the batch size during training has already been proposed in [1].
>
> **Reply to Weaknesses 1 and 2:** You are right that ''large batch sizes and small learning rates reduce test accuracy'' are well known, but did you know that the quantity $\eta C/\sqrt{b}$ contributes to the smoothing of the objective function?  This is a novel result we have uncovered. We admit that our algorithm is very similar to previous work [1] and that there is no novelty there, including numerical experiments. However, **the essence of our paper is smoothing by stochastic noise in SGD**, and the proposed algorithm is secondary. The question of why the methods of previous work [1] works well should not be able to be explained theoretically without an empirical reason: decreasing the learning rate improves performance. Our paper clarifies this theoretically from a completely different perspective from previous study [1]. We provide theoretical (Sections 3 and 5) and experimental (Sections 4 and 5) support for the commonplace technique of decreasing the learning rate and increasing the batch size by implicit graduated optimization. **This corroboration is our main contribution and novelty.**
>
> In addition, from Proposition 5.1, the decay rate of the degree of smoothing, $\gamma$, must satisfy $\gamma \in [0.5, 1)$ to guarantee convergence of the graduated optimization algorithm. Since we can guarantee that the degree of smoothing of the objective function is determined by $\delta=\eta C/\sqrt{b}$, by introducing graduated optimization, the optimal decay rate of the degree of smoothing immediately leads to the optimal decay and increase rate of the learning rate and batch size. Thus, we see that if the learning rate is to be reduced, it must be limited to a maximum of 0.5x in a single decay, and if the batch size is to be increased, it must be limited to a maximum of 4x in a single increase. This is a useful finding that cannot be obtained from previous studies.
> Let us emphasize again that **our greatest contribution is the connection between smoothing of the function by stochastic noise in SGD and graduated optimization.** This has never been done before and the results obtained are novel.
>
> ____
>
> **Weakness 3:** All methods achieved approximately 60\% in Figure 4. However, by comparing the results reported in the existing papers [1,2], 60\% appears to be too low. Thus, the reviewer is wondering if the results are reliable.
>
> **Reply to Weakness 3:** Reported in paper [1] is the training of ImageNet with ResNet50 and Inception-ResNet-v2. Our experiments were trained with ResNet34, so the results are not directly comparable. In the paper [2], ResNet18 is used, but the optimizer is SGD with momentum factor and weight decay. Since we use a simple SGD without momentum factor or weight decay to validate the theory, we cannot directly compare the results here either.
> Certainly, our results are not as good as the state-of-the-art in ImageNet classification, but we believe they are sufficient to validate our theory.
>
> ----
>
> **Reference**
>
> [1] Samuel et al., Don't Decay the Learning Rate, Increase the Batch Size, In ICLR 2018
>
> [2] He et al., Deep Residual Learning for Image Recognition, In CVPR 2016
>
> ---
>
> We deeply appreciate your careful reading of our paper. We would like to correct the typo appropriately.
> We specifically refuted the reviewers' concerns about novelty and differentiation from previous studies. We welcome further replies if you still have concerns.
> If you now have gone through our rebuttal, and you find that our paper is worthy of acceptance, please raise the rating score.

---

> > ### Comment · Reviewer_kxU4 · 2024-11-21
> >
> > The reviewer thanks the authors for their response.
> >
> > The reviewer carefully read the authors' responses and the comments of other reviewers and understood that the main contribution of this paper is finding the relationship between graduated optimization and SGD rather than proposing a new method.
> > However, the reviewer still believes that this finding is not very surprising and insufficient for acceptance.
> > Thus, the reviewer would like to keep the score.

---

### Official Review · Reviewer_k496 · 2024-11-03

**Soundness:** 3
**Presentation:** 4
**Contribution:** 2
**Rating:** 5
**Confidence:** 3

**Summary:**

This paper proposes the degree of smoothing notion in stochastic gradient descent and studies its relation with sharpness and generalization. From the proposed notion along with empirical studies, the paper observes that controlling the batch size and learning rate affects the degree of smoothness and therefore proposes a graduated optimization algorithm to gradually decrease the degree of smoothing by increasing the batch size and increasing the learning rate.

**Strengths:**

1. It is an interesting observation to view the update of SGD as smoothing.
2. The proposed degree of smoothing offers another intuitive explanation for decreasing learning rate and increasing batch size along the way of optimization and establishes its connection to graduated optimization.

**Weaknesses:**

1. The proposed degree of smoothing is somewhat obvious and simple, falling directly out of the variance/noise assumption of mini-batch SGD. Its correlation with concepts like sharpness is also straightforward because their definitions are somewhat similar already, with sharpness measuring the discrepancy of the function $f$ w.r.t. some $\delta$ neighborhood while the degree of smoothness the discrepancy of gradient $\nabla f$ w.r.t. some noisy disturbance $\omega$.
2. The numerical result is not quite informative as the effect of decreasing the learning rate or increasing the batch size has been studied and verified in previous optimization and learning theories like mini-batch SGD and sharpness-aware optimization.

**Questions:**

Is there any new insight/advantage the degree of smoothing offers other than decreasing the learning rate or increasing the batch size?

---

> ### Author Response · Authors · 2024-11-17
>
> **Reply to Weakness 1:** We think you are right. What is the weak point of our paper?
>
> **Reply to Weakness 2:** Yes, you are right, our numerical experiments are ultimately a common technique and there is nothing novel in them. As you are well aware, our novelty is that by bringing in the framework of graduated optimization, we have shown that the common techniques of decreasing learning rate and increasing batch size have theoretical significance.
>
> **Reply to Question:**
> 1. The degree of smoothing is correlated with generalization performance and may be useful for a better understanding of loss landscape and generalization performance.
>
> 2. The degree of smoothing is also correlated with Sharpness. Thanks to the degree of smoothing, we know that Sharpness is also correlated with the learning rate and batch size, since we have guaranteed the relationship between the degree of smoothing, the learning rate, and the batch size ($\delta=\eta C/\sqrt{b}$). These were observed experimentally [Keskar et al., 2017, Andriushchenko et al., 2023], but the reasons for this were not clear. This is the finding obtained by introducing the degree of smoothing.
>
> 3. For deriving the degree of smoothing, it is important to know what probability distribution the stochastic noise $\boldsymbol{\omega}$ follows. From the probability distribution requirement for function smoothing (see Reply to Question 2 of Reviewer KvGj), if the stochastic noise $\boldsymbol{\omega}$ in the SGD follows a light-tailed probability distribution, then the objective function can be considered to be smoothed by the noise $\boldsymbol{\omega}$. Conversely, if $\boldsymbol{\omega}$ follows a heavy-tailed distribution, smoothing of the function by $\boldsymbol{\omega}$ may not achieved and is outside the scope of our theoretical results.
> This provides theoretical support for the fact that SGD does not work when the noise $\boldsymbol{\omega}$ follows a heavy-tailed distribution. That is, when $\boldsymbol{\omega}$ follows a heavy-tailed distribution, the smoothing by noise $\boldsymbol{\omega}$ in SGD is not properly achieved, and SGD cannot avoid local solutions. It would be an interesting future work to see how it affects the theory of smoothing by noise when the noise $\boldsymbol{\omega}$ follows a heavy-tailed distribution in DNN training.
>
> We deeply appreciate your careful reading of our paper. We welcome further replies if you still have concerns. If you now have gone through our rebuttal, and you find that our paper is worthy of acceptance, please raise the rating score.

---

> > ### Comment · Reviewer_k496 · 2024-11-21
> >
> > Thank the authors for their response. While the reviewer acknowledges that it is insightful to observe the relation between smoothing and SGD and to connect this observation with graduated optimization, the paper's contribution does not seem substantial enough for the bar of ICLR in both theory and experiments. The reviewer would like to keep the original score.

---

### Official Review · Reviewer_NRH8 · 2024-11-04

**Soundness:** 3
**Presentation:** 3
**Contribution:** 2
**Rating:** 3
**Confidence:** 3

**Summary:**

This article discusses how Stochastic Gradient Descent (SGD), in its essence smoothens nonconvex functions while optimizing them theoretically analysis is provided here to show that the degree of smoothing ($\delta$) can be calculated using the formula $\delta= \eta C/\sqrt{b}$ where $\eta$ represents the learning rate and $C$ relates to variance while $b$ signifies the batch size. Additionally, it is theoretically and experimentally demonstrated that this smoothing effect clarifies findings in deep learning such as the reason behind poor generalization often observed with large batch sizes. The paper presents three contributions:
1. A mathematical model is offered to explain the smoothing effects of descent (SGDs).
2. There is a link between the level of smoothing and how the model performs overall; the best range for smoothing is between $0.1$ and $1.0$.
3. Introducing a graduated optimization technique that adjusts the level of smoothing by modifying the learning rate and batch size dynamically throughout the training process.

**Strengths:**

1. Introduce an innovative approach by showing that SGD’s inherent stochasticity can smooth nonconvex functions, it allows it to function as an implicit form of graduated optimization. This study leverages SGD’s existing stochasticity for the same purpose.
2. This paper offers a framework that explains the impact of learning rate adjustment and batch size variability on the level of smoothing in stochastic gradients. Its theoretical analysis is thorough and well supported by proofs. Clearly defined assumptions that provide a strong basis, for their assertions.

**Weaknesses:**

1. This work is constrained by the assumption that gradient noise follows a normal distribution, which will be expected for a broader category beyond normal distribution.
2. Analysis only focused on image classification tasks with CNN-based models.
3. The proof of convergence only applies to $\sigma$-nice functions, which is a restricted class of nonconvex functions.
4. Experiments are insufficient, mainly conducted on CIFAR100 with ResNet architectures, and no experiments on other domains beyond image classification.
5. Lack of discussion of computational overhead compared to standard SGD.
6. No discussion of how the method scales to relatively large models or datasets.

**Questions:**

Please refer to the weakness.

---

> ### Author Response · Authors · 2024-11-17
>
> **Reply to Weaknesses 1 and 2:** In the revised manuscript, we have extended the assumption of a Gaussian distribution; see the corrections in Section H and Definition 2.3.
> As you say, our theory of smoothing does not work unless the stochastic noise in SGD follows a light-tailed distribution (see Section H for details). Conversely, if $\boldsymbol{\omega}$ follows a heavy-tailed distribution, smoothing of the function by $\boldsymbol{\omega}$ may not achieved and is outside the scope of our theoretical results.
> It would be an interesting future work to see how it affects the theory of smoothing by noise when the noise $\boldsymbol{\omega}$ follows a heavy-tailed distribution in DNN training.
>
> **Reply to Weakness 3:** You are completely right. Currently, the $\sigma$-nice property is inevitably needed for convergence analysis of graduated optimization. It remains to be seen whether the empirical loss functions we consider actually satisfy $\sigma$-nice property, but we would like to add that some non-convex functions, such as Rastrigin's function (6) of any dimension, do satisfy $\sigma$-nice property.
>
> **Reply to Weakness 4:** This is currently being addressed. Please wait for our rebuttal.
>
> **Reply to Weakness 5:** As described in Section 5.2, a truly global optimization of a nonconvex function requires computational resources that can handle up to a full batch. Therefore, it will require more memory than vanilla SGD. However, the increase in memory usage is not a fatal flaw, as our experimental results (Figures 4-6) show that performance can be improved by simply increasing the batch size as much as is feasible for typical computing resources. Training time depends on the batch size, so depending on the initial batch size, it may take longer than vanilla SGD.
>
> **Reply to Weakness 6:** We provide numerical experiments on ImageNet and consider ImageNet to be a sufficiently large dataset. Since the experimental results for ImageNet are generally the same as those for CIFAR, we believe that our theory of smoothing and the graduated optimization algorithms that use it are effective for large models and datasets.
>
> We deeply appreciate your careful reading of our paper. We welcome further replies if you still have concerns. If you now have gone through our rebuttal, and you find that our paper is worthy of acceptance, please raise the rating score.

---

> > ### Author Response · Authors · 2024-11-25
> >
> > We deeply appreciate that you waited for our additional experiments.
> >
> > **Reply to Weakness 4:**
> > We have performed similar additional experiments on Sharpness and degree of smoothing on WideResNet-28-10, which is larger than ResNet18, and added the results to Figure 10 in Section G of the revised manuscript.
> > Figure 10 shows the excellent correlation between the degree of smoothing and Sharpness, as well as the clear relationship between the degree of smoothing and test accuracy, similar to the results for ResNet18.
> > Our additional experimentation eliminated some of the weaknesses pointed out by the reviewer.
> >
> > We welcome further replies if you still have concerns. If you now have gone through our rebuttal, and you find that our paper is worthy of acceptance, please raise the rating score.

---

> > > ### Comment · Reviewer_NRH8 · 2024-11-27
> > >
> > > Thank you for your replies and the substantial work you've done revisiting the document. I'm grateful for the explanations and the extra experiments you've included to tackle the issues I pointed out in my first review. The reviewer has the following concerns about maintaining the current score.
> > >
> > > **Weakness 3:** The reviewer gets that using $\sigma$-functions is important, for study purposes, but making your work more practically applicable would be a big plus if you expand the model to cover a wider range of non-linear functions too. Many real-life problems don't fit into the category, and seeing how well your approach works in such scenarios could offer valuable insights. Even though the reviewer agrees that empirical loss functions meet these criteria in practice exploring a range of function classes still represents a significant area, for future investigation.
> > >
> > > **Practical Implications:**
> > > 1. How do professionals decide on the level of smoothing or the right timing, for modifying the learning rate and batch size in situations? Can you offer advice or methods to tune these hyperparameters effectively in scenarios?
> > > 2. Are there any rules or tips that could help practitioners use your method without needing to adjust a lot of parameters?

---

> > > > ### Author Response · Authors · 2024-11-29
> > > >
> > > > **Reply to Weakness 3:**
> > > > In the definition of the smoothed function, suppose $u$ follows a standard normal distribution. In this case, we can show that the cross entropy loss, a commonly used empirical loss function, is a 1-nice function.
> > > >
> > > > Let $\boldsymbol{x}_i \in \mathbb{R}^d$ $(i \in [n])$ be $i$-th training data, $\boldsymbol{y}_i \in \mathbb{R}^c$ be $i$-th label (one-hot vector), and $f(\boldsymbol{x}_i)$ be $i$-th output of the model, where $d$ is the number of model parameters, $c$ is the number of class, and $n$ is the number of training data. Assume that the output elements $f(\boldsymbol{x}_i)^{(j)} \in \mathbb{R}$ $(j\in[c])$ is normalized to $(0,1]$ by sigmoid or ReLU.
> > > > In this case, the cross entropy loss can be expressed as
> > > > \begin{align*}
> > > > L := \frac{1}{n}\sum\_{i\in[n]}L\_i, \text{where }L_i := -\log f(\boldsymbol{x}\_i)^{(y\_i^{\text{hot}})},
> > > > \end{align*}
> > > > where $y\_i^{\text{hot}}$ is an index with element 1 of label $\boldsymbol{y}_i$, and $f(\boldsymbol{x}\_i)^{(y\_i^{\text{hot}})} \in \mathbb{R}$ is an element of $f(\boldsymbol{x}_i)$ corresponding to that index $y\_i^{\text{hot}}$.
> > > >
> > > > Therefore, we can consider the following function:
> > > > \begin{align*}
> > > > g(x) := - \log x \ (0<x\leq 1).
> > > > \end{align*}
> > > > Recall the definition of the smoothed function with Gaussian $u$,
> > > > \begin{align*}
> > > > \hat{g}\_{\delta}(x) := \mathbb{E}_{u \sim N(0;1)} [g(x-\delta u)].
> > > > \end{align*}
> > > > From the Taylor expansion, we have
> > > > \begin{align*}
> > > > -\log(x-\delta u) \approx -\log x + \frac{\delta u}{x} + \frac{\delta^2 u^2}{2x^2}.
> > > > \end{align*}
> > > > Hence,
> > > > \begin{align*}
> > > > \hat{g}\_{\delta}(x)
> > > > := \mathbb{E}\left[ -\log(x-\delta u) \right]
> > > > = -\log x + \frac{\delta^2}{2x^2},
> > > > \end{align*}
> > > > where we use $\mathbb{E}[u]=0, \mathbb{E}[u^2]=1$.
> > > >
> > > > First, both $g$ and $\hat{g}\_{\delta}$ have a global minimum at $x=1$, i.e., $x^\star = 1$ and $x\_{\delta}^\star=1$ for all $\delta$. Thus, the first condition of $\sigma$-nice is satisfied.
> > > >
> > > > Next, since $g''(x) = \frac{1}{x^2} \geq 1$ and $\hat{g}\_{\delta}''(x) = \frac{1}{x^2} + \frac{3\delta^2}{x^4} \geq 1$ hold, both $g$ and $\hat{g}_{\delta}$ are 1-strongly convex. Thus, the second condition of $\sigma$-nice is also satisfied.
> > > >
> > > > Therefore, the function $g$ i.e., $L_i$ is 1-nice function. Since cross entropy loss $L$ is the average of $n$ functions $L_i$, $L$ is also 1-nice function.
> > > >
> > > > The deadline has passed so we can't share the pdf, but we can add this important discussion to the revised manuscript.
> > > > There are some assumptions, but we believe that this result is very insightful and you are right, it is a big plus for our paper.
> > > > This new result is a result of your kind and informative comments. We really deeply appreciate your careful peer review and active discussion.
> > > >
> > > > **Reply to Practical Implications 1:**
> > > > Theoretically, the timing to update parameters is when the training loss stops decreasing for training with the current parameters.
> > > >
> > > > Consider the case where training is determined to be 200 epochs, as in our experiment.
> > > > According to our algorithm, the number of times to modify the parameters is $M-1$.
> > > > Since $M:=\log_{\gamma}\alpha_0 \epsilon$, a smaller threshold $\epsilon$ can be achieved by making $M$ larger. However, a larger $M$ would reduce the number of iterations at each stage and thus would not allow for sufficient optimization. Therefore, in this case, there is an optimal value for $M$. Indeed, we performed a grid search among $M=\[2,4,5,8,10,20\]$ and found that $M=4,5$ achieved the lowest training loss function value and the highest test accuracy.
> > > >
> > > > **Reply to Practical Implications 2:**
> > > > For example, the degree of smoothing must decrease from a sufficiently large value, so it is useful to use a sufficiently small initial batch size $b_1$, such as 8 or 16. If the initial batch size is large, such as 512, the smoothing will not be sufficient and the algorithm will not be able to achieve its full potential.
> > > >
> > > > In addition, it is useful to start with commonly used general parameters and update the learning rate and/or batch size when the losses stop decreasing. In particular, increasing only the batch size and fixed learning rate will not slow down the optimization process because the learning rate does not have to be small, and increasing the batch will also increase the computational speed, allowing the training to be completed faster than maintaining a small batch size throughout the training.
> > > >
> > > > Once again, We would like to express our deepest gratitude.
> > > > If you now have gone through our reply, and you find that our paper is worthy of acceptance, please raise the rating score.

---

> > > > > ### Comment · Reviewer_NRH8 · 2024-12-03
> > > > >
> > > > > Thanks for the authors' detailed response. The reviewer has carefully read their response. However, the major reason for not accepting to enhance the score is because even though the paper offers important theoretical contributions, it fails to adequately address the issues of generalisability of the findings to several other non-linear functions thus leaving some critical practical issues for future research instead of addressing them in this submission.

---

> > > > > > ### Author Response · Authors · 2024-12-03
> > > > > >
> > > > > > Thanks for your reply.
> > > > > >
> > > > > > We have responded to your reasonable comment by showing that the cross-entropy loss, which is most commonly used in classification tasks, is the 1-nice function. From the same discussion, we can also show that the mean square error, which is often used in regression tasks, is a 2-nice function.
> > > > > >
> > > > > > You are right that we have not extended the definition of the $\sigma$-nice function, and a more relaxed version of it may indeed be an important issue.
> > > > > > However, for the applications of the $\sigma$-nice function and graduated optimization approach in machine learning that are most important to us and to you, aren’t these new results enough?
> > > > > > We consider these results to be valuable advances for both machine learning and graduated optimization, and therefore our paper is worthy of acceptance.
> > > > > >
> > > > > > Time is short, but we welcome your replies.

---

### Official Review · Reviewer_KvGj · 2024-11-04

**Soundness:** 3
**Presentation:** 3
**Contribution:** 1
**Rating:** 3
**Confidence:** 4

**Summary:**

This paper studies the convergence of stochastic gradient descent (SGD) in the context of nonconvex optimization. The authors aimed to show that the gradients help the objective by smoothing it through the noise injected by sampling functions. The claim that SGD smoothes the objective is shown by assuming that the gradients are distributed according to isotropic Gaussian distribution, which I find to be a trivial result. Moreover, since the work is written from the perspective of giving a new theory for SGD specifically, I find this to be very misleading. The authors also present experiments on CIFAR100 to study the numerical properties related to generalization such as sharpness, which serve as a secondary contribution. Next, the authors propose a new method for $\sigma$-nice functions that runs gradient descent on a smoothed objective with varying parameters and they explain why the method works. Finally, the authors run several variations of SGD on training ResNet-34 on ImageNet to show that increasing batch size helps SGD converge.

**Strengths:**

1. I think a theory for SGD and an explanation why noise helps to train neural networks is highly desired. It is a great topic and if the results were good, I'd have considered this an important contribution.
2. The numerical evaluations are reasonable.

**Weaknesses:**

1. My main concern about this work is the unrealistic assumption that the noise from sampling gradients follows Gaussian distribution with identity covariance matrix and variance that does not change over the course of training. What's worse, this assumption is not stated as clearly as other assumptions, instead it's introduced in the text and a couple of references are given to experimental papers that justify normality of the gradients. Those papers, however, do not show that gradients have exactly the same distribution throughout training. It's also never discussed in the paper why the assumption should hold or what happens if it doesn't. And what we should expect here, in contrast, is that the noise level changes every iteration and its variance is a random variable that depends on the iterates and previously sampled gradients.
2. Since the gradient noise is assumed to be exactly gaussian and consntant, the paper fails to deliver what the abstract promises, namely to "show that stochastic noise in stochastic gradient descent (SGD) has the effect of smoothing the objective function", because the authors essentially *assume* that the noise smoothes the objective. I usually refrain from calling a result trivial.
3. Since the results in this work assume Gaussian noise, it means that prior papers on injecting noise inside gradients immediately apply to SGD in this setting. However, there is no comparison to related work on this topic, such as Orvieto et al. "Anticorrelated Noise Injection for Improved Generalization" and Vardhan & Stich, (2021). The latter paper is only mentioned in passing as showing that noise helps escape saddle points, but the authors do not explain what novelty their paper has to offer.

## Minor
The abstract says that "The graduated optimization approach is a heuristic method", which is not true since it has already been studied in the work of Hazan et al. (2016). It's particularly inappropriate since the authors use the same assumption of $f$ being $\sigma$-nice
The objective function $f$ is not properly introduced before being used in the introduction
"noise smooths" -> "noise smoothes"
"diffusion models (Sohl-Dickstein et al., 2015; Ho et al., 2020; Song et al., 2021a; Rombach et al., 2022), which are currently state-of-the-art generative models, implicitly use the techniques of graduated optimization". This seems like a very streched example, diffusion models are injecting noise in the image or latent space and for reasons very different from minimizing a nonconvex function. I suggest the authors remove this statement or give a reference where it is shown that there exists a function implicitly minimized by image denoising
Lemmas 2.1 and 2.2 are introduced with no context, which leads to an unnatural flow when reading the paper. Perhaps the discussion that follows them could be put prior to stating the lemmas.
Broken citation: "Harshvardhan" should have been "Harsh Vardhan"
Line 420, "is nonnegative constant" -> "is a nonnegative constant"

**Questions:**

1. In what sense do the authors "show" that noise in SGD helps? I see no theory for this, it all seems to follow from assuming Gaussian distribution of gradient noise and prior literature.
2. Can the assumption on Gaussian noise be removed?
3. It appears to me that log scale in Figure 2 in x-axis is actually not helpful as most growth seems to happen for larger values on the x-axis, especially in Figure 2 (B). Can you show us the figure with the x-axis not scaled logarithmically?

---

> ### Author Response · Authors · 2024-11-17
>
> **Reply to Weakness 1:** You are correct that the noise level changes every iteration, but its level always has $\frac{C}{\sqrt{b}}$ as an upper bound, as shown by Lemma 2.1. In Section 3, we use this and introduce $\boldsymbol{\omega}_t = \frac{\eta C}{\sqrt{b}}\boldsymbol{u}_t$. Thus, we did not assume that the noise level does not change during the training process, but simply took an upper bound on the noise level for the analysis. Also, we have changed our assertion about the distribution of $\boldsymbol{\omega}_t$. See our revised manuscript and our reply to Question 2 for details.
>
> **Reply to Weakness 2:** The reply to Weakness 1 should clear up the misunderstanding that we assumed that the gradient noise is constant. Please see the reply to Question 2 for the distributions of $\boldsymbol{\omega}_t$. Since these concerns should have been addressed, the discussion in Section 3 is valid and we have accomplished what we promised in the abstract.
>
> **Reply to Weakness 3:** We are deeply grateful to you for presenting us with relevant studies to cite. We would like to add it appropriately.
>
> ---
> **Reply to Question 1:** The previous study by Kleinberg et al. does not mention the probability distribution of the gradient noise that you pointed out, so it only suggests that the stochastic noise in the SGD smoothes the objective function, and the main contribution of their paper is the convergence analysis based on that assumption. As noted in Section 7, we have extended Kleinberg et al.'s argument, formally defining function smoothing and showing what factors determine the degree of smoothing. This is our major contribution, and the experimental comparison of the degree of smoothing with Sharpness and generalization performance and the introduction of graduated optimization are also novel contributions not found in previous studies. **Our greatest contribution is the connection between smoothing of the function by stochastic noise in SGD and graduated optimization.**
> This has never been done before and the results obtained are novel.
>
> ---
> **Reply to Question 2:** Yes. We have made two modifications to our paper to address the weakness you pointed out.
>
> 1. **We change the definition of smoothing of the function (see Section H and Definition 2.3).**
> We have added a discussion of what probability distribution a random variable should follow for function smoothing. As a result, we found that smoothing of the function is possible if the random variable does not follow a Gaussian distribution, but follows a light-tailed distribution.
> Motivated by this argument, we changed the probability distribution that the random variable $\boldsymbol{u}$ follows in the definition of function smoothing from Gaussian distribution to light-tailed distribution.
>
> 2. **We measure the distribution of the stochastic noise $\boldsymbol{\omega}_t$ (see Section H.2).**
> To see what distribution the noise $\boldsymbol{\omega}_t$ follows, we sampled 1000 $\boldsymbol{\omega}_t$ at 10,000 steps to see what distribution each element follows. As a result, we are not sure if $\boldsymbol{\omega}_t$ follows a Gaussian distribution, but it clearly follows a light-tailed distribution. $\boldsymbol{\omega}_t$ seems to follow a Gaussian-like distribution, so further verification is needed to confirm whether it really follows a Gaussian distribution. However, our theory of smoothing does not require that $\boldsymbol{\omega}_t$ strictly follows a Gaussian distribution, but only that $\boldsymbol{\omega}_t$ follows a light-tailed distribution.
>
> These two modifications eliminate the need for the Gaussian assumption that the reviewers were concerned about and assure the soundness of the theory of smoothing (see Section 3).
> Your helpful remarks have made our paper even more robust. We are deeply grateful to you.
>
> ---
> **Reply to Question 3:** Yes, we have included a figure without the x-axis on a logarithmic scale at the end of the Appendix of the revised manuscript. Please see Figure 13.
>
> ---
> **Reply to minor comment on our abstract:** You are completely right. We apologize for seeming to downplay the theoretical results of Hazan et al.
>
> **Reply to minor comment on diffusion models:** Diffusion models, and score-based generative models in particular, optimize a multi-peaked, nonconvex probability density function that the training data are assumed to follow. Introducing noise in the image contributes to smoothing the probability density function.
> The technique of introducing noise of different sizes in sequence is a graduated optimization, although the procedure was introduced from a completely different perspective of avoiding low data density regions.
> Therefore, our explanation is correct. Please see for example, Section 3.2 of "Generative Modeling by Estimating Gradients of the Data Distribution" (NeurIPS2019).
>
> **Reply to other minor comment:** We deeply appreciate your remarks on the definitions and typos. We would like to correct them appropriately.

---

> > ### Comment · Reviewer_KvGj · 2024-11-18
> >
> > > Thus, we did not assume that the noise level does not change during the training process, but simply took an upper bound on the noise level for the analysis.
> >
> > I can see that you updated the paper and now it doesn't assume a fixed distribution at any point $x$. Unfortunately, this breaks how you use the definition of $\sigma$-nice function, in particular Definition 5.1 (ii) has nothing to do with what Hazan et al. (2016) were writing about. In their paper, the distribution is fixed, but your updated Definition 2.1 has an undefined light-tailed distribution $\mathcal{L}$. As far as I can see, this makes the statement of Theorem 5.1 invalid.

---

> ### Author Response · Authors · 2024-11-20
>
> We have modified the definition of smoothing, but the point that there exists a fixed batch size $b$ such that $\boldsymbol{\omega}_t$ follows a common distribution independent of time $t$ has not been changed from the original manuscript.  An exact description is provided below.
>
> We would like to find $\boldsymbol{\omega}_t$, such that
> \begin{align*}
> \mathbb{E}\_{\xi\_t}\left[ \Vert \boldsymbol{\omega}_t \Vert\right] \leq \frac{C}{\sqrt{b}}.
> \end{align*}
> The $\boldsymbol{\omega}\_t$ for which this equation is satisfied can be expressed as $\boldsymbol{\omega}\_t = \frac{C}{\sqrt{b}}\boldsymbol{u}\_t$, where $\mathbb{E}\_{\xi\_t}\left[ \Vert \boldsymbol{u}\_t \Vert \right] \leq 1$.
> Then, based on the experimental observation (see Section H.2 and several previous studies [Zhang et al., 2020; Kunstner et al., 2023]) that $\boldsymbol{\omega}_t$ follows a light-tailed distribution, we assume $\boldsymbol{\omega}_t \sim \hat{\mathcal{L}}$, where $\hat{\mathcal{L}}$ is a light-tailed distribution.
> From $\boldsymbol{\omega}\_t = \frac{C}{\sqrt{b}}\boldsymbol{u}\_t$, $\boldsymbol{u}\_t \sim \mathcal{L}$ holds, where $\mathcal{L}$ is a light tailed-distribution and is a scaled version of $\hat{\mathcal{L}}$.
>
> Thus, we use that for fixed batch size, $\boldsymbol{\omega}\_t$ follows the same distribution $\hat{\mathcal{L}}$ for any time $t$ and $\boldsymbol{u}_t$ follows the same distribution $\mathcal{L}$ for any time $t$. We do not believe this is a fatally strong assumption. To clarify these, the description in Section 3 has been modified.
>
> ---
>
> Also, the reviewer doubts the soundness of Theorem 5.1. In Theorem 5.1, $\delta_m$, i.e., the batch size $b\_m$, is predetermined, and the batch size $b\_m$ is constant throughout the optimization of $\hat{f}\_{\delta\_m}$. Note that we do not assume that the distribution of $\boldsymbol{\omega}\_t$ always follows $\hat{\mathcal{L}}$ for any given batch size. If the batch size changes, of course the distribution that $\boldsymbol{\omega}_t$ follows will change, and the distribution that $\boldsymbol{u}\_t$ follows will change accordingly. Therefore, there is no problem with the statement of Theorem 5.1.

---

> > ### Comment · Reviewer_KvGj · 2024-11-20
> >
> > >Thus, we did not assume that the noise level does not change during the training process
> >
> > This is what you said in your first response
> >
> > >Thus, we use that for fixed batch size, $\omega_t$ follows the same distribution $\hat{\mathcal{L}}$ for any time $t$ and $u_t$ follows the same distribution $\mathcal{L}$ for any time $t$
> >
> > This is what you said in your second response.
> >
> > Don't these two statements contradict each other?
> >
> > > Therefore, there is no problem with the statement of Theorem 5.1.
> >
> > I did not understand your explanation, so let me clarify what I think is problematic. The issue that I see with the theorem is in how you use the definition of $\sigma$-nice function. In particular, Definition 5.1 (ii) states that $\hat{f}_{\delta_m}$ is $\sigma$-strongly convex over
> >
> > $N(x_{\delta_m}^{*} ; 3\delta_{m})$. When providing the definition, you simply cite the work of Hazan et al. (2016) as if it was used like that there, but Definition 3.1 in (Hazan et al., 2016) uses noise sampled from a unit ball $\mathbb{B}$. In other words, their work does not have proofs for light-tailed distributions with bounded variance and their theory cannot be applied directly in your work.

---

> > > ### Author Response · Authors · 2024-11-21
> > >
> > > We deeply appreciate your prompt reply.
> > >
> > > ---
> > >
> > > > Don't these two statements contradict each other?
> > >
> > > No, they don't. However our statements may not have been clear. We apologize to you for our unclear reply.
> > > In the training process with Algorithm 1, the distribution that $\boldsymbol{\omega}\_t$ follows changes as $m$ changes (more precisely, as batch size $b_m$ changes), however, for a fixed $m$ (i.e., fixed batch size $b_m$), the distribution that $\boldsymbol{\omega}\_t$ follows does not change for any $t$.
> > >
> > > ---
> > >
> > > >The issue that I see with the theorem is in how you use the definition of $\sigma$-nice function. In particular, Definition 5.1 (ii) states that $\hat{f}\_{\delta\_m}$ is $\sigma$-strongly convex over $N(\boldsymbol{x}\_{\delta\_m}^\star; 3\delta\_m)$. When providing the definition, you simply cite the work of Hazan et al. (2016) as if it was used like that there, but Definition 3.1 in (Hazan et al., 2016) uses noise sampled from a unit ball $\mathbb{B}$. In other words, their work does not have proofs for light-tailed distributions with bounded variance and their theory cannot be applied directly in your work.
> > >
> > > The difference of the definition of the smoothed function $\hat{f}\_{\delta}$ between Hazan et al. and our paper is the distribution that the random variable $\boldsymbol{u}$ follows.
> > > As you say, Hazan et al. defined that the random variable $\boldsymbol{u}$ follows a uniform distribution from the unit Euclidean ball $\mathbb{B}$ (note that in their analysis of $\sigma$-nice property, they used a Gaussian distribution).
> > > In contrast, we define the random variable $\boldsymbol{u}$ to follow any light-tailed distribution.
> > > The light-tailed distribution is a probability distribution that has a thinner tail than an exponential distribution.
> > > We can show that the uniform distribution is light-tailed distribution (see revised Section H.1).
> > > Thus our Definition 2.1 contains Definition 4.1 of Hazan et al. and does not conflict with it. Therefore, there is no problem with the statements of Definition 5.1 and Theorem 5.1.
> > >
> > > We have added this explanation just below Definition 2.1 and Section H.1.

---

> > > > ### Comment · Reviewer_KvGj · 2024-11-21
> > > >
> > > > > Thus our Definition 2.1 contains Definition 4.1 of Hazan et al.
> > > >
> > > > Exactly, your definition is more general and it allows for an arbitrary light-tailed distribution, and that's why it doesn't make sense to assume that $\hat{f}_{\delta_m}$ is $\sigma$-strongly convex (which is assumed in your Defition 5.1 (iI)). Cosinder, for instance, the case where $u\equiv 0$, which is possible since constant distribution is also light-tailed with bounded variance. If that's the case, then there is no difference between the smoothed function and the original one, which means that you are assuming the original objective to be strongly convex in that specific case.
> > > >
> > > > More generaly, the strong convexity constant should of $\hat{f}_{\delta_m}$ should depend on the amount of noise that you have. This is in contrast to the work of Hazan et al. who work with a fixed distribution and can, therefore, assume strong convexity to be constant regardless of the current iterate.

---

> > > > > ### Author Response · Authors · 2024-11-22
> > > > >
> > > > > Indeed, we must exclude the case $P\left( \lbrace \omega \in \Omega \colon \boldsymbol{u}(\omega) = \boldsymbol{0} \rbrace \right) = 1$, since smoothing is not achieved if the distribution $\mathcal{L}$ returns $\boldsymbol{u}=\boldsymbol{0}$ with probability 1.
> > > > > Otherwise, since the smoothed function $\hat{f}\_{\delta\_m}$ takes the expected value of $\boldsymbol{u}$, our definition of smoothing is fine.
> > > > >
> > > > > ---
> > > > >
> > > > > You are right, it was not realistic to assume that smoothed function $\hat{f}\_{\delta\_m}$ is $\sigma$-strongly convex for any $\delta_m$. To avoid this problem, we modified $\hat{f}\_{\delta\_m}$ so that its strong convexity parameter $\sigma_m$ depends on the degree of smoothing $\delta_m$ (see revised Definition 5.1, Theorems 5.1 and 5.2, and Section I).
> > > > >
> > > > > We deeply appreciate your valuable comments.

---

> > > > > > ### Author Response · Authors · 2024-11-25
> > > > > >
> > > > > > Our revised manuscript has added full results for the distribution of $\boldsymbol{\omega}$ (see Figure 14 in Section I).
> > > > > > Your thorough peer review and dedicated discussions have made our paper more robust.
> > > > > > We are deeply grateful. If you now have gone through our rebuttal, and you find that our paper is worthy of acceptance, please raise the rating score.

---

### Official Review · Reviewer_4qQd · 2024-11-06

**Soundness:** 3
**Presentation:** 3
**Contribution:** 3
**Rating:** 6
**Confidence:** 3

**Summary:**

The paper presents a heuristic approach for solving nonconvex optimization problems by combining a smoothing technique. The authors demonstrate that stochastic gradient noise impacts the smoothing of the objective function, with the extent of this effect determined by three factors: the learning rate, batch size, and the variance of the stochastic gradient. Building on these insights, the authors introduce a new graduated optimization method. Theoretical analysis and numerical results confirm the effectiveness of the proposed method.

**Strengths:**

The paper offers a novel perspective on the smoothing effect of stochastic gradient descent (SGD) and its implications for optimizing nonconvex functions.

The connection between smoothing by SGD and generalization performance is a contribution to this field. The correlation between the degree of smoothing, sharpness of the objective function, and generalization performance is convincingly shown, enhancing the credibility of the theoretical insights.

**Weaknesses:**

While the experiments with ResNets on CIFAR100 provide valuable insights, they may not fully generalize to other types of neural networks or more complex datasets.

A more comprehensive discussion on the practical implementation of the proposed implicit graduated optimization algorithm would further enhance its applicability and understanding.

**Questions:**

How do different optimizer variants (e.g., Adam, RMSprop) impact the smoothing effect observed with SGD?

What strategies can practitioners use to effectively set the initial values and decay rates for learning rate and batch size to maximize the advantages of implicit graduated optimization?

Could this framework be extended to analyze optimization in graph neural networks or manifold learning?

What computational trade-offs might be associated with implementing the proposed algorithm, such as increased training time or memory usage?

**Details Of Ethics Concerns:**

N.A.

---

> ### Author Response · Authors · 2024-11-14
>
> **Weakness 1:** While the experiments with ResNets on CIFAR100 provide valuable insights, they may not fully generalize to other types of neural networks or more complex datasets.
>
> **Reply to Weakness 1:** This is currently being addressed.
> Please wait for our rebuttal.
>
> ---
>
> **Weakness 2 and Question 2:** A more comprehensive discussion on the practical implementation of the proposed implicit graduated optimization algorithm would further enhance its applicability and understanding. What strategies can practitioners use to effectively set the initial values and decay rates for learning rate and batch size to maximize the advantages of implicit graduated optimization?
>
> **Reply to Weakness 2 and Question 2:** Thanks for your great comment and question! Since we can guarantee that the degree of smoothing of the objective function is determined by $\delta=\eta C/\sqrt{b}$, by introducing graduated optimization, the optimal decay rate of the degree of smoothing immediately leads to the optimal decay and increase rate of the learning rate and batch size.
> From Proposition 5.1, the decay rate of the degree of smoothing, $\gamma$, must satisfy $\gamma \in [0.5, 1)$ to guarantee convergence of the graduated optimization algorithm. Thus, we see that, if the learning rate is to be reduced, it must be limited to a maximum of 0.5x in a single decay, and if the batch size is to be increased, it must be limited to a maximum of 4x in a single increase. This may not be a major finding for practitioners, but we emphasize that it is a novel contribution.
>
> ---
>
> **Question 1:** How do different optimizer variants (e.g., Adam, RMSprop) impact the smoothing effect observed with SGD?
>
> **Reply to Question 1:** You are right, extending the argument for SGD in Section 3 to Adam and RMSProp is a natural extension: by considering the difference between the search direction of Adam and that of GD in the same way, we can derive the degree of smoothing due to stochastic noise that Adam has. It is expected that the momentum factor $\beta_1, \beta_2$, and other hyperparameters will be included, which may provide new insights into Adam's behavior, just as our paper provided new insights into SGD's behavior. Furthermore, it may be possible to construct an implicit graduated optimization algorithm that exploits this property. We believe that these are very important future work derived from our results.
>
> ---
>
> **Question 3:** Could this framework be extended to analyze optimization in graph neural networks or manifold learning?
>
> **Reply to Question 3:** Our framework, from smoothing the objective function with stochastic noise to implicit graduated optimization, is useful for analyzing all problem settings that minimize the nonconvex empirical loss function.
> Note, however, that the stochastic noise must follow a light-tailed distribution, such as a normal or uniform distribution.
>
> ---
>
> **Question 4:** What computational trade-offs might be associated with implementing the proposed algorithm, such as increased training time or memory usage?
>
> **Reply to Question 4:** As described in Section 5.2, a truly global optimization of a nonconvex function by implicit graduated optimization requires computational resources that can handle up to a full batch. Therefore, it will require more memory than vanilla SGD. However, the increase in memory usage is not a fatal flaw, as our experimental results (Figures 4-6) show that performance can be improved by simply increasing the batch size as much as is feasible for typical computing resources. Training time depends on the batch size, so depending on the initial batch size, it may take longer than vanilla SGD.
> Finally, we would like to add that **our main contribution is to theoretically support the advantages of practical techniques such as the proposed algorithm through a graduated optimization framework.**
>
> ---
>
> We deeply appreciate the careful peer review.
> If you still have any concerns or comments, by all means reply!
> If you think it is worthy of acceptance through our rebuttal, please raise your rating score.

---

> ### Author Response · Authors · 2024-11-25
>
> We deeply appreciate that you waited for our additional experiments.
>
> **Reply to Weakness 1:** We have performed similar additional experiments on Sharpness and degree of smoothing on WideResNet-28-10, which is larger than ResNet18, and added the results to Figure 10 in Section G of the revised manuscript.
> Figure 10 shows the excellent correlation between the degree of smoothing and Sharpness, as well as the clear relationship between the degree of smoothing and test accuracy, similar to the results for ResNet18.
> Our additional experiments eliminated the weakness pointed out by the reviewer.
>
> If you still have any concerns or comments, by all means reply! If you think it is worthy of acceptance through our rebuttal, please raise your rating score.

---

### Author Response · Authors · 2024-12-04

We deeply appreciate the reviewers' careful review and active replying during the discussion period.

To assist the Chairs in their final decision, we summarize the reviewers and our arguments here.

The reviewers who recommended reject mainly raised the following issues
1. the SGD's gradient noise does not follow a Gaussian distribution or is not sure if it does (by Reviewer KvGj).
2. the analysis is only valid for $\sigma$-nice functions (by Reviewer NRH8).

To address this issue, we made the following additions and corrections.
1. **We modified the definition of smoothed function based on experimental observations and theory (see Definition 2.1 and Section H).**
This eliminates the assumption in our theory that the gradient noise of SGD follows a Gaussian distribution, and instead relaxes the assumption that it follows a light-tailed distribution. Issue 1 has been resolved, and Reviewer KvGj seems to be satisfied because the replying has ceased.

2. **We have shown that the cross entropy loss and the mean square error are 1-nice and 2-nice functions, respectively (see the reply to Reviewer NRH8).** The essence of issue 2 is that our theory is not valid for practical loss functions because the $\sigma$-nice function is a function that satisfies special conditions, and we do not know whether practical loss functions satisfy those conditions.
Cross-entropy loss and mean squared error are loss functions commonly used in classification and regression tasks, respectively. Therefore, the fact that these are $\sigma$-nice functions also eliminates issue 2.
Reviewer NRH8 does not seem to be satisfied because we have not significantly extended the definition of the $\sigma$-nice function to more nonlinear functions. However, for the applications of the $\sigma$-nice function and graduated optimization approach in machine learning that are most important to us and to all reviewers, we believe that our additional results are sufficient and that this additional results more than raises the value of our paper.

The other reviewers' arguments are summarized below.
- Reviewer 4qQd appreciates the theoretical and experimental novelty and value of our paper and seem to agree with the acceptance of our paper.
- Reviewers k496 and kxU4, while acknowledging the correctness of our theory and the sufficiency of our experiments, are unwilling to accept our results as not surprising.

All of the above shows that our paper has no shortcomings other than the perceived lack of impact on the two reviewers and provides theoretically and experimentally correct results.

Our main contributions are listed below.
1. We show that the degree of smoothing $\delta$ provided by SGD's stochastic noise depends on the quantity $\delta = \frac{\eta C}{\sqrt{b}}$, where $\eta$ is the learning rate, $b$ is the batch size, and $C^2$ is the variance of the stochastic gradient.
2. We experimentally show the relationship between the degree of smoothing, Sharpness, and generalization performance.
3. By bringing in graduated optimization, we showed that the use of decaying learning rate and increasing batch size actually contributed to the search for the global optimal solution.
4. An implicit graduated optimization algorithm is proposed and analyzed.
5. (Additional.) We showed that cross entropy loss and mean squared error, which are very common in machine learning, are $\sigma$-nice functions.

We consider any of these contributions to be novel and surprising.
- Contributions 1 and 2 provides new insights into the relationship between the landscape of the loss function and generalization performance from a completely different perspective from previous studies, such as Sharpness.
- Contribution 3 provides theoretical support for the general method of decreasing learning rate and increasing batch size.
- Contribution 4 is the first ever attempt at implicit graduated optimization without an explicit smoothing operation.
- Contribution 5 allows all previous theoretical and experimental findings on graduated optimization to be introduced into machine learning through the $\sigma$-nice function.

We are sorry to impose the hard work on the Chairs, but for these reasons, we believe that this paper is worthy of acceptance.
In addition to the rating scores, we ask that you consider our arguments and those of the reviewers before making a final decision.
Finally, our paper is more correct and more valuable thanks to the reviewers' valuable remarks. We sincerely appreciate the hard work of the reviewers.

---

### Meta-Review · Area_Chair_oU8K · 2024-12-05

**Metareview:**

The paper examines the smoothing effects of SGD on nonconvex functions, as influenced by the learning rate, batch size, and the variance of the stochastic gradient. While the reviewers found the exploration of the relationship between SGD parameters and smoothing a relevant topic, concerns were raised about the significance of the findings, arguing that they may offer limited novelty given existing knowledge about SGD. Additionally, some reviewers described the assumptions as overly restrictive and argued that the experimental work was not sufficiently persuasive. The authors are encouraged to incorporate the important feedback given by the knowledgeable reviewers.

**Additional Comments On Reviewer Discussion:**

Strong objections to acceptance were raised on account of the issues outlined above, which remained unresolved despite the discussions.

---

### Decision · Program_Chairs · 2025-01-22

Reject